# Clone detection for business process models

Mahdi Saeedi Nikoo[1], Önder Babur[1,2] and Mark van den Brand[1]

[1] Department of Mathematics and Computer Science, Eindhoven University of Technology, Eindhoven, The Netherlands
[2] Information Technology Group, Wageningen University & Research, Wageningen, The Netherlands



Corresponding author
Mahdi Saeedi Nikoo,
m.saeedi.nikoo@tue.nl

## ABSTRACT

Models are key in software engineering, especially with the rise of model-driven software engineering. One such use of modeling is in business process modeling, where models are used to represent processes in enterprises. As the number of these process models grow in repositories, it leads to an increasing management and maintenance cost. Clone detection is a means that may provide various benefits such as repository management, data prepossessing, filtering, refactoring, and process family detection. In model clone detection, highly similar model fragments are mined from larger model repositories. In this study, we have extended SAMOS (Statistical Analysis of Models) framework for clone detection of business process models. The framework has been developed to support different types of analytics on models, including clone detection. We present the underlying techniques utilized in the framework, as well as our approach in extending the framework. We perform three experimental evaluations to demonstrate the effectiveness of our approach. We first compare our tool against the Apromore toolset for a pairwise model similarity using a synthetic model mutation dataset. As indicated by the results, SAMOS seems to outperform Apromore in the coverage of the metrics in pairwise similarity of models. Later, we do a comparative analysis of the tools on model clone detection using a dataset derived from the SAP Reference Model Collection. In this case, the results show a better precision for Apromore, while a higher recall measure for SAMOS. Finally, we show the additional capabilities of our approach for different model scoping styles through another set of experimental evaluations.

## INTRODUCTION

Model-driven engineering (MDE) is being adopted and used by increasingly more software-driven organizations and enterprises. As models are central artifacts in the MDE paradigm, their analysis also has gained more importance over time. With the proliferation of models, duplication in software and model repositories has become more evident (*Koschke, 2008*; *Roy, Cordy & Koschke, 2009*). This is also the case for business process model repositories in industry (*Weber et al., 2011*).

Clones in model repositories may originate from reusing existing models, where a process designer generates new models by copy-pasting and modifying existing models

(*Rattan, Bhatia & Singh, 2013*). This can also result from co-existing a set of model variants belonging to the same process family. As an example, consider an airport check-in process every passenger has to go through and the possible variations in the steps (*e.g.,* online *vs.* in-person check-in). There may be variants of the process with similarities which would manifest itself in the form of clones. In both mentioned cases, the developed models would show a high degree of lexical similarity. These duplications may lead to several issues. For instance, inconsistencies may occur across models if clones are modified separately by different stakeholders. Also, as duplications increase, repositories grow in size, which also incurs extra maintenance and management costs.

There may be various reasons behind finding similarity in models. Foremost, clones may imply scenarios for refactoring and quality assurance in MDE/DSL ecosystems (*Deissenboeck et al., 2010*). Also, since such ecosystems in larger enterprises may consist of several DSLs, clone detection across different DSLs and their versions offers a good potential for empirical studies on them and on their evolution (*Tairas & Cabot, 2011*). Furthermore, clone detection in model repositories could provide enterprises with benefits such as better repository management, exploration, data preprocessing, filtering, and empirical studies regarding their origin, distribution, and so on (*Uba et al., 2011*). There could also be other uses, *e.g.,* for analyzing conformance to a given reference model, or for standardization such as generation of a standard process model from a set of similar models (*La Rosa et al., 2015*). Finally, we might use clone detection to find process model variants which can be considered a process family in process line engineering discipline (*Boffoli et al., 2012*). Although outside the scope of this article, in our research, we're interested in the latter direction. Using clone detection, our overarching goal is to identify potential business process families in large model repositories, which would subsequently be used for building product lines based on them and managing their variability (*Cognini et al., 2018*).

Our clone detection technique is implemented based on the BPMN 2.0 notation as it is the de-facto standard for business process modeling, but the technique could also be adapted for other process modeling notations. To achieve this goal, we started to look into the available tooling for business process model clone detection. However, our search led to finding only one publicly available tool (Apromore) which we could use in our comparative evaluations. We found another relevant tool which is introduced in *Skouradaki et al. (2016)*. Although it is partially similar to our work, in the sense that both approaches include identifying similarities between models, however, the scope and goal of the works are different. The approach by *Skouradaki et al. (2016)* is about detecting structural patterns among a set of BPMN 2.0 models. Also, it does not consider textual similarities of model elements, which means the detected pattern fragments are not similar from a content point of view. A detected pattern among a given set of model fragments does not necessarily mean they are clones. Therefore we did not include this in our study. Our technique is implemented as an extension to Statistical Analysis of Models (SAMOS) framework (*Babur, Cleophas & van den Brand, 2019*).

SAMOS is a framework for large-scale analysis of models based on information retrieval and machine learning techniques. The underlying techniques used in the framework are

generic to graph-based modeling languages, and have been successfully applied to *e.g.*, Ecore metamodels (*Babur, Cleophas & van den Brand, 2019*), feature models (*Babur, Cleophas & van den Brand, 2018*), statecharts (*Wille et al., 2018*) and industrial domain-specific models (*Babur et al., 2020*). As business process models also use a graph-based structure for workflow definition, we found the framework to be suitable for implementing our approach. In this work, we have made the following contributions:

- We present a technique for business process model clone detection which is realized as an extension to SAMOS.
- We present a set of applied adaptations and refinements on the base framework SAMOS.
- We provide quantitative comparative evaluations of our refined approach with other existing approaches in terms of accuracy and effectiveness using various metrics.

The results show that, by extending SAMOS, as a generic framework for model clone detection, we are able to get promising accuracy in results for clone detection of business process models.

The rest of the article is organized as follows: In "BPMN Clones", we provide an introduction to BPMN model clone detection. In "Preliminaries", a brief review of background knowledge relevant to our study is provided. In "Methodology", we present our clone detection approach. "Experimental Evaluation", explains multiple experimental evaluations used to compare and evaluate our approach against the existing tools. "Overall Discussion", provides a discussion about the possible future work. "Related Work", discusses some major related works in the domain. Finally, "Conclusion and Future Work" concludes the article.

## BPMN CLONES

In this study, our aim is to find similar fragments in BPMN models. There can be several reasons for applying clone detection. One is that clones can result in scenarios for quality assurance and refactoring in MDE ecosystems. In the case of refactoring, for instance, it can be used to extract methods by grouping together similar code fragments in different classes in the code (*Arcelli Fontana et al., 2013*). Also, model repositories in industry, publicly available ones, and the ones used for research, could leverage clone detection for different purposes such as repository management, exploration, data filtering and processing, and empirical studies. Lastly, clone detection can be used for detecting plagiarism and possibly in grading assignments in educational courses about modeling. A similar idea is followed in the source code domain (*Prechelt, Malpohl & Philippsen, 2002*).

The following classification scheme is widely used in the literature for model clone detection as a standard, which was initially developed for Simulink models (*Alalfi et al., 2012*):

- **Type-I (exact) model clones**: Identical model fragments except for variations in visual presentation, layout, and formatting.

- **Type-II (blind renamed, or consistently renamed) model clones**: Structurally identical model fragments except for variations in labels, values, types, visual presentation, layout, and formatting.
- **Type-III (near-miss) clones**: Model fragments with further modifications, such as changes in position or connection with respect to other model fragments and small additions or removals of blocks or lines in addition to variations in labels, values, types, visual presentation, layout, and formatting.
- **Type-IV (semantic) clones**: Model fragments with different structure but equivalent or similar behavior.

SAMOS has adopted a slightly different classification scheme than the above, which is an adaptation of the scheme presented by *Störrle (2015)* for UML models.

- **Type-A** duplicate model fragments except formatting, layout, or internal identifiers. Also, any cosmetic changes to name (lower/upper case, snake/camel case, and similar minor changes).
- **Type-B** duplicate model fragments with limited amount of changes to types, names, attributes, and few addition and/or removal of elements. Also, may include multiple syntactic/semantic changes to the names such as typos, synonyms, semantically relevant terms.
- **Type-C** duplicate model fragments with significant amount of change/additions/ removals of types, names, attributes, and parts.
- **Type-D** semantically equivalent or similar model fragments with different structure and content.

In this study, we use the classification as defined in SAMOS, with one special consideration. In the latter scheme, Type-A clones correspond to Type-I (exact) clones in the standard scheme, while Type-B clones are closely related to Type-III clones in the standard scheme. As Type-C clones include the same type of changes as Type-B clones, only in a higher magnitude, we do not distinguish between the two classes in this study. These both are the type of clones that emerge as a result of copy-pasting existing models in model repositories and applying arbitrary changes to them. In this study, we do not consider Type-D clones as they introduce bigger challenges to tackle.

## PRELIMINARIES

In this section, first, we provide an overview of business process modeling and briefly discuss the used notations. Next, we introduce the two model clone detector tools, namely SAMOS and Apromore, which are used in this work. SAMOS is used as the base framework in our approach, while Apromore is used as the only available tool that offers process model clone detection. Later in our experiments, we compare SAMOS to Apromore to provide an evaluation of our approach. An overall discussion of the underlying techniques used in the two tools is presented in this section.

## Business process models

Several modeling notations have been introduced so far to model the business processes, including UML Activity Diagrams (*Fowler, 2004*), Event-driven Process Chains (EPCs) (*Mendling, 2008*), Business Process Modeling Notation (BPMN) (*Object Management Group (OMG), 2013*), Yet Another Workflow Language (YAWL) (*Van Der Aalst & Ter Hofstede, 2005*), among others. Among these, BPMN is the most commonly used modeling notation.

EPC notation was introduced in 1992 with the purpose of creating a language for a clear representation and documentation of business process models (*Keller, Scheer & Nüttgens, 1992*). EPC models consist of core and extended elements. The core elements provide a basic definition of a process model. Only the core elements are formalized and well-documented. Extended elements were added to add some organizational structure and data flow to the process models, which could not be specified by the core elements. The core elements of EPC models consist of functions, events, and connectors. Functions model the activities of a business process, while events indicate occurrence of activities, which affects the following process flow. Connectors are used to describe how functions and events are connected, and may be one of AND (logical conjunction), OR (inclusive disjunction), or XOR (exclusive disjunction) types. Connectors, also have a splitting or joining behavior, which can start with functions or events. The connection between process elements is *via* control-flow arcs.

BPMN (*Object Management Group (OMG), 2013*) is a modeling language standardized by the Object Management Group (OMG) for describing the functional behavior of a business process. The main goal of BPMN is to allow designing visual models of business or organizational processes in a standardized notation that are understandable by all business stakeholders. BPMN defines a Business Process Diagram (BPD), which is using flowcharting techniques for creating graphical models of business process operations. Before the standardization of the BPMN notation, EPC models were widely used as a de-facto standard in industry. With more expressive power and tool support behind it, BPMN has become more widely used in the industry.

Accordingly, a business process model consists of a network of graphical objects, which are mainly activities (*i.e.*, tasks) with flow controls that define their order of execution (*White, 2004*). The concrete syntax of BPMN consists of four basic categories of elements: Flow Objects, Connecting Objects, Swimlanes, and Data.

- Flow objects form the overall process workflow. The three main flow objects are events, activities, and gateways. Events serve as a trigger, initiating a start point, intermediate step, or end point of a process. Activities illustrate a specific task performed by a person or system. Activities can also be in various forms such as ones that occur once, occur multiple times, or occur if a specific set of conditions are met. Gateways represent decision points that determine the directions to take along a process.
- Connecting objects are used to connect process elements together. Sequence flow is the main connecting object that is used to connect and show order of flow objects.

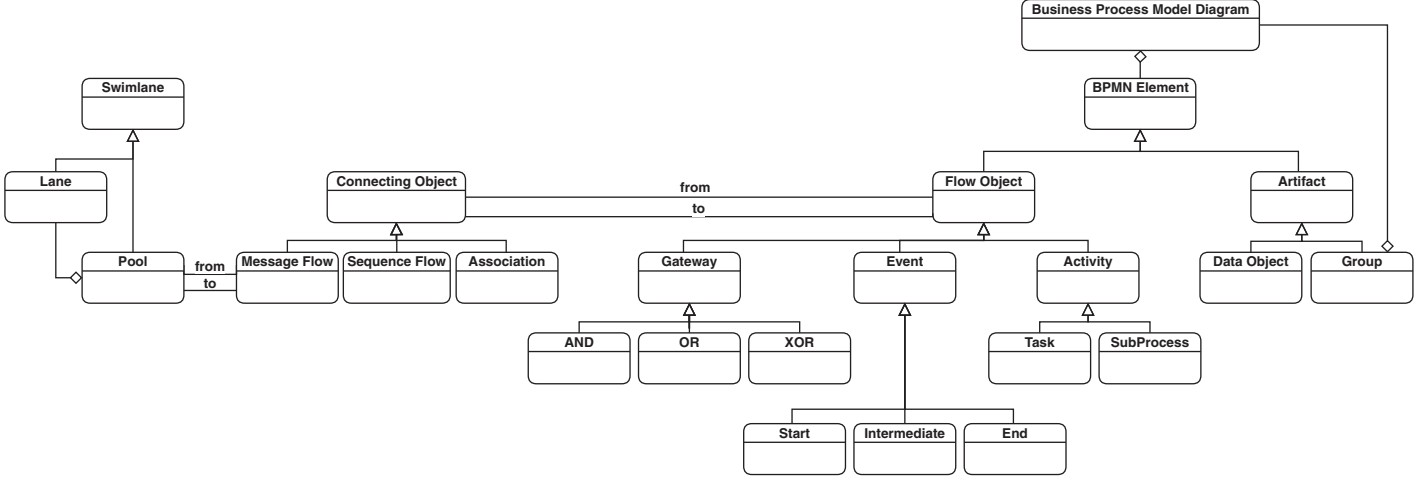

**Figure 1 A simplified version of BPMN metamodel.**

- Swimlanes represent participants of a business process. There are two types of swimlanes: pools and lanes. A pool represent an entire department such as marketing, while lanes encompass the activities for a specific role such as sales engineer in the department.
- Data objects are used to represent a certain type of data or information consumed or produced during the execution of a process.

A simplified version of the BPMN metamodel is presented in Fig. 1 that includes the subset of the language concepts that cover the mentioned four basic categories of elements. The full specification of BPMN notation can be found in (*Object Management Group (OMG), 2013*). There are also other elements such as Event types that can be assigned to Event nodes, or various types of Tasks such as Manual or Service Task which are not shown in Fig. 1 as core concepts in BPMN process modeling, which are also considered in our study. This is elaborated more under "Methodology". In this study, we consider the represented elements in the depicted metamodel, which constitute the essential elements for defining a process model using BPMN notation. There is also a support for choreography modeling in BPMN, but as we do not consider this type of modeling, the corresponding elements are excluded from our study.

## SAMOS model clone detector

SAMOS (Statistical Analysis of Models) is a framework for large-scale analysis of models (*Babur, Cleophas & van den Brand, 2019*). It treats models as documents in the Information Retrieval (IR) terminology. Starting with a feature extraction phase on these models, further analysis and clustering can be performed. Features can be as simple as element names in models, or more complex structures including fragments of the model graph structure such as n-grams (*Manning & Schutze, 1999*). SAMOS calculates a vector space model (VSM), and applies weighting schemes and natural language processing

**Table 1** An example of term incidence matrix representation of Shakespeare's plays (excerpt from *Manning, Raghavan & Schtze (2008)*).

|  | Anthony and cleopatra | Julius caesar | The tempest | Hamlet | Othello | Macbeth | ... |
|---|---|---|---|---|---|---|---|
| Anthony | 1 | 1 | 0 | 0 | 0 | 1 | |
| Brutus | 1 | 1 | 0 | 1 | 0 | 0 | |
| Caesar | 1 | 1 | 0 | 1 | 1 | 1 | |
| Calpurnia | 0 | 1 | 0 | 0 | 0 | 0 | |
| Cleopatra | 1 | 0 | 0 | 0 | 0 | 0 | |
| ... | | | | | | | |

(NLP) techniques. It can also perform sophisticated statistical calculations such as distance measuring and clustering *via* the R statistical computing language.

The underlying techniques used in SAMOS are mainly inspired from information retrieval (IR) and machine learning (ML) domains. IR deals with indexing, analyzing, searching and comparing different forms of contents from document repositories, particularly textual information (*Manning, Raghavan & Schtze, 2008*). As a starting point, the collected documents are indexed *via* some unit of representation (vocabulary), which can include a bag of words (all words or selected ones). As an alternative, n-grams, which originate from computational linguistics can be used, which are more complex constructs. N-grams represent a linear encoding of (text) structure, *e.g.*, "Julius Caesar" as a single entity instead of identifying each word separately.

VSM can be used to implement an index construction. A VSM has the following main components: (1) a vector representation of the vocabulary occurrence or frequency of a document. (2) Optionally zones, *e.g.*, 'author' or 'title', (3) weighting schemes such as inverse document frequency (IDF), and zone weights, (4) NLP techniques for treating compound terms, detecting synonyms or semantical similarities.

To give an example of a VSM, Table 1 shows an excerpt from *Manning, Raghavan & Schtze (2008)* that presents a simplistic representation of Shakespeare's play. The vocabulary covers some important terms, and the vector space is filled with the incidence (*i.e.*, not frequency) of these terms in the respective plays.

As shown, using the VSM we can transform a document into an $n$-dimensional vector, and as a result in an $m \times n$ matrix for $m$ documents. With the VSM, we can define distance using *e.g.*, Euclidean, Manhattan, or cosine between two given vectors. Following the incidence matrix in the given example, a simple dot product of the VSM in Table 1, would result in the $m \times m$ pair-wise distance matrix, where for instance "Anthony and Cleopatra"·"Julius Caesar"=*3*, "Julius Caesar"·"Hamlet"=*2* and "Julius Caesar"·"The Tempest"=*0*.

With such a distance matrix, we may then apply an unsupervised ML technique called clustering to identify similar groups of documents (*Manning, Raghavan & Schtze, 2008*; *Jain & Dubes, 1988*). K-means and hierarchical clustering are the two well-known techniques for this. With k-means, the aim is to find cluster centers and minimize the residual sum of (square of) distances of the assigned points in each cluster. With

hierarchical clustering, there is no assumption on the number of clusters, and rather a nested tree structure (*dendrogram*) is built from the data points, which represents proximity and potential clusters.

## Apromore approximate model clone detector

*La Rosa et al. (2015)* present an approach for approximate clone detection for business process models. The ultimate goal in their work is to retrieve clusters of approximate clones for standardization and refactoring into shared subprocesses. The approach is geared towards detecting approximate clones of fragments that originate from copy-pasting followed by independent modifications applied to the copied fragments.

According to the authors, the presented approach in the article applies to directed graphs with labeled nodes, which can be applied to different process modeling notations such as EPC and BPMN. They initially define an abstract representation of process models based on labeled graphs. The definition is limited to control-flow elements of process models, namely events and tasks, which are labeled, and gateways, which are unlabeled. However, as mentioned in their article (*La Rosa et al., 2015*), it can be extended to be applied to non-control-flow elements such as object and roles.

For comparing pairs of process models, they first define distance measures for doing comparison in node level. For calculating distance between labeled nodes, they adopt combination of two techniques, namely syntactic and semantic distance. In syntactic distance, labels are treated as strings and a string-edit distance technique is applied. For semantic distance, however, semantic relatedness such as synonymy of labels is considered. They use a WordNet library for the semantic measure. In the case of comparing gateway nodes which are unlabeled nodes, they adopt another approach named context distance, where they compare the preceding and succeeding labeled nodes of gateway nodes.

Having the distance measure defined for nodes, they then define a distance measure for process graphs based on the known notion of graph-edit distance (*Messmer, 1995*). The graph-edit distance between two graphs is the minimum number of operations required to transform one graph to the other. To lower the computational complexity of measuring graph-edit distance, they adopt a greedy heuristic, as the original algorithm is NP-complete (*Dijkman, Dumas & Garca-Bañuelos, 2009*). The defined process graph and graph-edit distance are assumed to work for graphs with labeled nodes and unlabeled edges, but the authors claim they can be extended to work with edge labels (*Dijkman et al., 2011a*; *La Rosa et al., 2013*).

They have three main considerations for the approximate clones. One is to avoid containment clones, *i.e.*, avoid calling two fragments an approximate clone only because of one being contained in the other. The second is that given the goal of refactoring identified approximate clones into subprocesses which follows a call-and-return semantics, the detected clones need to be in the form of single-entry, single-exit (SESE) fragments. In short, an SESE fragment has exactly two boundary nodes: one entry and one exit. The third one is to avoid trivial clones, *i.e.*, fragments with single activity which are not suitable to be a subprocess.

The proposed approach is based on the two techniques: Refined Process Structure Tree (RPST) (*Vanhatalo, Völzer & Koehler, 2009*), and RPSDAG (*Dumas et al., 2013*). The RPST, which is a parsing technique, is used to compute a unique tree representing the hierarchy of SESE fragments retrieved from the given process model. RPSDAG is an index structure that is used to hold up the union of RPSTs of process models. RPSDAG, which is built incrementally by adding new process models, allows for exact clone identification in a collection of process models, also shows the containment hierarchy of model fragments to be used during clustering.

The mentioned techniques are utilized for the approximate clone detection among the extracted SESE fragments. To that end, a distance matrix is generated for the collection of fragments using different distance measures, including the graph-edit distance between a pair of fragments. In order to detect approximate clones, two different clustering algorithms namely Density-Based Spatial Clustering of Applications with Noise (DBSCAN) and Hierarchical Agglomerate Clustering (HAC) are used. The presented approach is implemented as a plugin of the Apromore platform (*La Rosa et al., 2011*).

## METHODOLOGY

In this section, we present our methodology for clone detection in BPMN models. For that, we show how we utilize existing set of techniques provided by a generic model analytics framework (*i.e.*, SAMOS) with the goal of conducting clone detection for BPMN models. We present the set of customizations we apply to the framework to achieve this goal. Each section presents a relevant technique or aspect in the framework and provides a more in-depth discussion on the extensions and the changes applied to the framework.

### Using and extending SAMOS for business process model clone detection

Our approach in this article is based on SAMOS (*Babur, Cleophas & van den Brand, 2019*). The framework provides a generic (notation-neutral) set of techniques that can be used for clone detection of any given graph-based modeling notation. SAMOS has already been applied for metamodel clone detection. Given that, for using the framework for a specific modeling notation, parts of the applied techniques need to be specialized based on the target notation. For BPMN notation, these customizations are specifically applied to the scoping for similarity checking, feature extraction, and vertex and n-gram comparison stages in the framework. The main customizations are given below and their detailed explanations are provided in the following sub-sections:

- Scoping for similarity checking: This concept is related to any type of modularity inherent in the given notation. As the framework is language-neutral, we defined a set of scoping concepts specific to BPMN notation. Scoping concept is defined in "Scoping for Similarity Checking" and elaborated more in "Experiment 3: Clone Detection With Lane and Subprocess Scoping".

- Feature extraction: Feature extraction in its naive meaning encompasses the extraction of all the existing data and their local relations from the language metamodel as

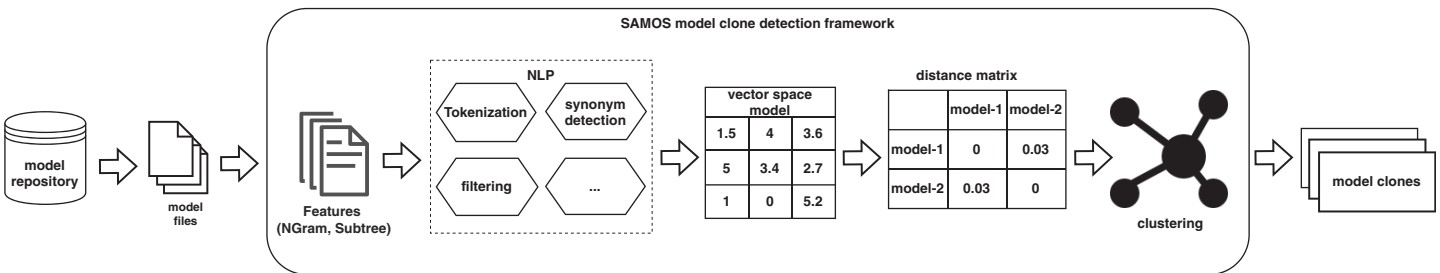

**Figure 2 SAMOS clone detection workflow.**

information units for clone detection. However, in practice this is not feasable for a given notation, as every piece of information extracted from metamodel is not of the same value for our purpose. We applied BPMN-specific customizations for this stage as elaborated in "Extracting Model Element Information".

- Vertex and n-gram comparison: We applied another set of customizations on how vertex and n-grams are compared. A different similarity calculation formula with a set of adjustments considering BPMN elements were defined, which are explained in more detail in "Vertex and N-gram Comparison".

In the following sections, we explain different aspects and functionalities of the framework. Figure 2 shows an overview of the key steps taken in SAMOS for clone detection.

## Scoping for similarity checking

An important criterion in similarity measurement in clone detection is the input for the similarity algorithm. *Schoknecht et al. (2017)* propose a classification in *Schoknecht et al. (2017)* on the input based on 4 categories: (1) similarity between (sets of) model elements, (2) similarity between sub-graphs, (3) similarity between two models, and (4) similarity between sets of models. The mentioned classification corresponds to the notion of scoping in SAMOS.

By scoping, we can define the granularity in which we want to do feature extraction from the models. By default, the extraction process extracts data about all the model elements. For BPMN models, we may have several scopes, but in this study we define the following scopes: the whole model, SESE, Subprocess and Lane. The SESE scoping is not currently implemented in SAMOS, however, we utilize Apromore to indirectly apply this scoping to BPMN models. This is shown in "Experiment 2: Model Clone Detection on Asset Management Dataset" in more detail. These scopes are chosen as they constitute main elements in the containment tree of a process model which contain other elements. That is why they make a natural slicing of models into logically grouped set of elements that can potentially be clone fragments in models. The scoping is set at the beginning of each run of the clone detection (*Babur, Cleophas & van den Brand, 2019*). The user thus

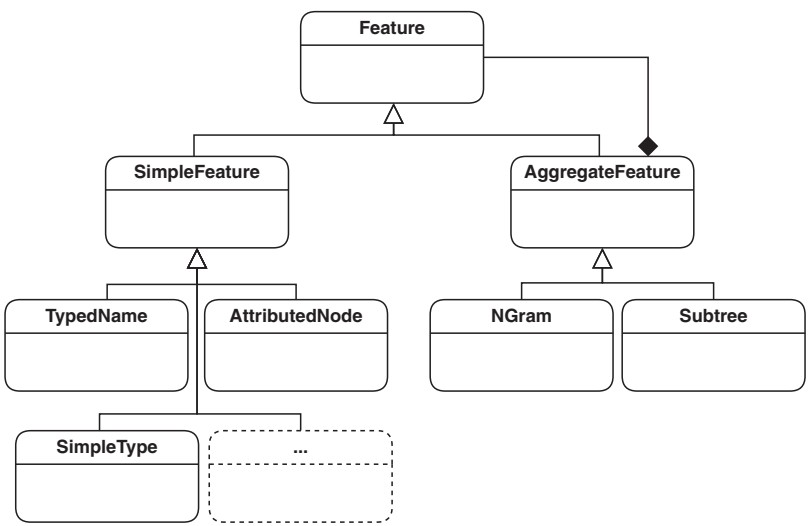

**Figure 3 Feature hierarchy used in SAMOS: simple features for encoding vertices and edges, and aggregate features for representing structure.**

can choose one of the mentioned scopes besides the default scoping, which is at the model level, consisting of all elements in a model.

## Extracting model element information

Features basically define the unit of information extracted from models that represent essential information for the type of processing done on models. The basic definition of a feature in SAMOS has previously been the so called type-name pair, which would map to a vertex in the graph of the model. Such a pair is enough to encode the domain-specific type information, such as EClass, and the name, such as Book, of a model. To improve clone detection, we need to extend this definition to include the attributes in model elements (*e.g.*, whether an EClass is abstract) and cardinalities, *e.g.*, of EReferences. Capturing edge information such as containment is also helpful in the comparison. The current feature hierarchy in SAMOS which covers the mentioned model information is represented in Fig. 3. The mentioned extensions for feature extration in the feature hierarchy are shown as (1) AttributedNode, which holds all the information of a vertex covering its domain-specific type, name, type, and attributes as key-value pairs; and (2) SimpleType, which indicates whether an edge is of contaitnment or supertype (*i.e.*, superclasses as named in EMF). The mentioned feature types are subclasses of Simple-Feature class and represent non-composite, stand-alone features.

There are some key points to mention regarding the feature extraction component of our tool. These are mainly about how features are extracted, what criteria are considered for the extraction process and the processing applied on the feature names.

### *Metamodel-based feature extraction*

In our implementation, we use the BPMN 2.0 Ecore-based metamodel as a reference and use the Eclipse plugin BPMN2 Modeler (*Eclipse, 2021*) API to extract model information.

**Listing 1** Selected bigram extractions from the model in Fig. 4.

```
v₁ = {type:StartEvent, name:every_ten_minute, lane: import_order_from_marketplace_to_erp,  parellelMultiple:false,
  eventDefinition:timer_event_definition_impl, isInterrupting:true},

v₂ = {type:UserTask, name:collect_all_order_from_marketplace, lane: import_order_from_marketplace_to_erp,
  completionQuantity:1, isForCompensation:false, startQuantity:1},

v₃ = {type:Task, name:check_order_datum, lane: import_order_from_marketplace_to_erp, subprocess:handle_order,
  completionQuantity:1, isForCompensation:false, startQuantity:1},

v₄ = {type:ExclusiveGateway, name:datum_correct_?, lane: import_order_from_marketplace_to_erp, subprocess:handle_order,
  gatewayDirection:diverging}

v₅ = {type:SequenceFlow, name:yes,isImmediate:false}
```

### Feature selection

Features in SAMOS make up a representation of a given model that are used as the only source of information in the next phases of clone detection. In a naive strategy for the feature extraction, we would transform the whole model (*i.e.*, covering all the elements in the metamodel) into a set of features based on our feature definition. Due to the big size of the BPMN 2.0 metamodel and the large number of artifacts defined in the notation, extracting all corresponding features from a model leads to a huge amount of information, a significant part of which is not much relevant for the clone detection. Therefore, we limit our implementation mainly to flow elements of the process graph and the relevant features based on that. The process graph oriented feature extraction is enriched with extra attributes which would otherwise be extracted as separate nodes from the model.

### Additional attributes for features

As our feature extraction mainly considers features related to flow elements in BPMN 2.0, by excluding other features we lose part of the information that may improve the accuracy of our similarity algorithm. Our approach to partly compensate for this information loss is to embed such information somehow in the selected extracted features. For instance, a feature for an *Event* element is extracted, but the *Event Definition* of that event is excluded based on the mentioned reasoning. However, we may add additional attributes to the extracted feature for Event to show the type of *Event Definition* of it. Similarly, there is no extracted feature for a subprocess, but it can be added as an attribute to all child elements of the subprocess to preserve such information. This can be seen in the examples in the Listing 1.

### Feature name normalization

We use a normalizing pass on all the names of the extracted elements to convert them into a uniform format composed of the lemmatized tokens with all white spaces trimmed and stop words removed. This results in a better treatment of the names in the vertex comparison algorithm introduced in "Vertex and N-gram Comparison"

To exemplify, Listing 1 shows the extracted features for the marked vertices in Fig. 4.

A naive feature extraction according to the full metamodel specification would normally result in much more features than the ones shown in Listing 1. However, we have applied

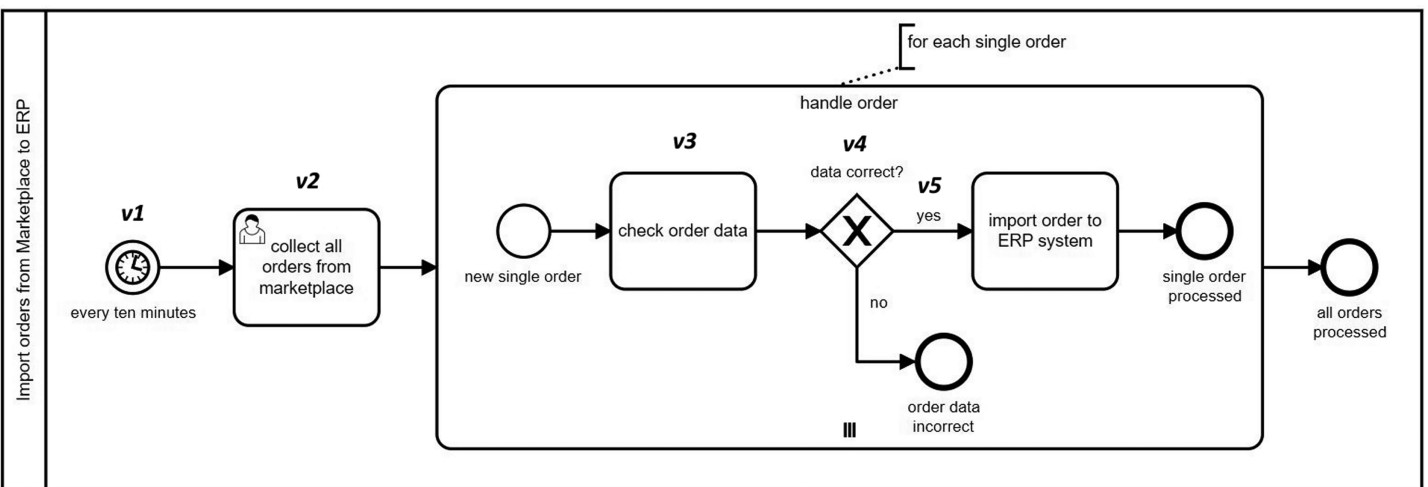

**Figure 4  A sample BPMN model for handling orders.**

modifications to the feature extraction component in our approach. There are basically two adjustments: The first one is to limit the number of features to the ones most essential for our clone detection goals. The second one is to enrich included features with additional attributes extracted from the excluded features (*i.e.*, removed as a result of the first adjustment). As an example, in $v_1$ there is an attribute named *eventDefinition* which would normally be a separate feature with its own set of attributes, but is injected as an attribute to *StartEvent*. The main reason behind this is to avoid the redundant information that does not play a significant role to distinguish between two given BPMN models in our initial attempt for BPMN clone detection. It is also important to note that every feature that we extract from a model has a part in the representation of the model as a whole as well as all the furthur processing based on the feature set afterwards. Therefore, cluttering the feature list with non-primary features has a negative impact on result accuracy of the tool.

## Encoding structure in n-grams

As we elaborate more in "Distance Measurement", there are different dimensions for similarity measurement of business process models. There are already multiple studies considering structural aspects in process model similarity measurement (see for example *La Rosa et al. (2013)* and *Sánchez-Charles et al. (2016)*). Our approach also relies on a (graph) structural context in similarity measurement. To that end, model fragments are extracted, which are encoded as features. SAMOS supports three different settings for encoding features. In our implementation so far, we are using unigram and n-gram settings and leave subtrees for future work.

- **Unigram**: model structure is ignored and nodes are used as-is (*Babur, Cleophas & van den Brand, 2016*);

- **N-gram**: model structure is encoded in linear chunks (*Babur & Cleophas, 2017*) (for $n > 1$);
- **Subtree**: model structure is encoded as fixed depth subtrees.

In the above-mentioned settings, unigrams correspond to SimpleFeatures in the conceptual feature hierarchy in Fig. 3. N-grams and subtrees (potentially with depth $n > 1$) are aggregated features containing multiple SimpleFeatures. According to a graph representation of a model, we can think of n-grams as n consecutively connected vertices. In order to represent the edge information, edges are incorporated as SimpleTypes in the n-gram encoding. The readers can refer to *Babur & Cleophas (2017)* for a more detailed discussion on graph traversal and n-gram extraction. Some bigrams ($n = 2$) from Fig. 4 are given below:

- $b_1 = (v_1, \text{outgoing}, v_2)$
- $b_2 = (v_4, \text{outgoing\_yes}, v_5)$

We make a modification to bigram extraction to decrease the repetition in the extracted features. This is due to the fact that in BPMN specification, connectors such as sequence and message flow are also flow elements, thus considering them as separate vertices results in encoding a fragment such as [*A*] *seqFlow* [*B*] as two separate features: *A seqFlow* and *seqFlow B*. The partial repetition here increases dramatically when there are tens of such fragments in the model. The result set thus has a negative effect in the VSM building, which propagates to the next steps in clone detection. For this reason, we consider BPMN connectors as edges. Bigram $b_2$ in the above list shows an example of how the encoding works. The middle part in bigram $b_2$ means there is an outgoing sequence flow labeled *yes* from $v_4$ to $v_5$.

## Vertex and n-gram comparison

As aggregate features consist of multiple Feature vertices, we first give a definition of the vertex comparison. Vertex comparison in SAMOS is defined as a multiplicative formula as given in Eq. (1), where nameSim holds the NLP-based similarity between the names, while typeSim and eTypeSim hold the similarity of the domain types and eTypes in the vertices, respectively. The last variable attrSim, defined in Eq. (2) is used for measuring similarity of attributes of model elements which are defined in the metamodel (for instance, gatewayDirection in a Gateway or isForCompensation in a Task element).

$$vSim(n_1, n_2) = nameSim(n_1, n_2) * typeSim(n_1, n_2) * eTypeSim(n_1, n_2)$$
$$* attrSim(n_1, n_2) \tag{1}$$

$$attrSim(n_1, n_2) = 1 - \frac{\# \; unmatched \; attributes \; between \; n_1 \; and \; n_2}{total \; \# \; attr. \; for \; that \; domain \; type} \tag{2}$$

In our implementation, we opt for an additive formula as shown in Eq. (3), instead of a multiplicative one, with the same variables as defined in Eq. (1). This change is made as we need to make multiple tweaks to the similarity measures in the formula, and we can do this

in a more controlled way with an additive formula. Another change is on the nameSim variable, whose effect in the formula is doubled, as we want to give somewhat more weight to element names in the comparison. The last change is about the omission of the eType parameter. eType is a classifier parameter used in Ecore metamodel elements. We exclude it in our similarity calculation, as it has a trivial effect in the formula compared to other parameters.

$$vSim(n_1, n_2) = 2 * nameSime(n_1, n_2) + typeSim(n_1, n_2) + attrSim(n_1, n_2) \tag{3}$$

The above formula may not be the optimal one, and we formulated it this way after some explorative iterations of running the tool and looking for the best results. We leave it as future work to improve this scheme. For N-gram comparison, we use the semi-relaxed formula (*Babur & Cleophas, 2017*), *i.e.*, given n-grams $\vec{v_1}$ and $\vec{v_2}$ with $2n - 1$ elements ($n$ vertices and $n - 1$ edges corresponding to $v_1^{1..2n-1}$ and $v_2^{1..2n-1}$), the n-gram similarity is:

$$nSim(\vec{v_1}, \vec{v_2}) = \frac{1 + |nonzero\ vSim\ matches\ between\ \vec{v_1}\ and\ \vec{v_2}|}{1 + (2n - 1)} \tag{4}$$

Considering the various possible BPMN model element compositions, there also needs to be additional workaround in the similarity measurement. Below, we provide the ones already present in SAMOS and the ones we add in the scope of this article:

- In SAMOS, in the case of non-matching types, the attributes are ignored altogether.
- SAMOS ignores edge-only matches such as A-relation-B *vs*. C-relation-D when A-C and B-D have a zero similarity. The relation in our case can be *e.g.*, an outgoing sequence flow from A to B.
- We give a lower weight to sequence flows, thus its similarity weight is halved.

SAMOS also supports NLP techniques in computing *nameSim*, including label normalization in case of mixed-casing, tokenization and compound word similarity, stemming and lemmatization, Levenshtein distance for typos, and Wordnet-based synonym checking.

### Additional customizations

In order to increase our tool's accuracy for distance calculation, we apply some additional minor customizations on the n-gram comparison component of SAMOS. For a better distance measurement, the goal is to increase the distance between more dissimilar models and decrease it for more similar ones. The applied changes are listed below:

- While comparing two bigrams, say AxB and CyD (A, B, C, D being vertices; x and y being edges), when there is no name similarity between the vertices, *i.e.*, A compared to C and B compared to D, then the edge similarity is set to 0.
- We always assume attribute similarity score of edges to be 1. This is because edges do not have any attributes in the bigram formulation in SAMOS.

- When the types of two vertices do not match, their corresponding name and attribute values are set to 0.
- The comparison algorithm has a different treatment when the compared vertex types belong to the same family. For instance, both User Task and Manual Task are subtypes of Task, thus there should be less distance between them when compared to a different type.
- If there is name match on one vertex only in bigram, the type, and attribute weights for that vertex, as well as the edge weight is lowered.
- Modifying a gateway with multiple incoming and/or outgoing connectors has a higher impact on the overall similarity score. A local weighting scheme is applied to adjust the similarity score based on the incoming/outgoing flow size of gateways.

## VSM calculation

SAMOS provides two modes for VSM calculation: linear and quadratic (all-pairs). Linear VSM counts the exact feature occurrences, meaning it computes the total frequencies of features, but this mode is not powerful enough to calculate synonyms or fine-grained differences in features such as attributes and types. However, quadratic VSM compares each feature occurrence in a model against the entire feature set of all models and does a better treatment of fine-grained differences. Regarding their performance, linear mode is much faster and cheaper as a single pass is enough to build the VSM, while quadratic mode leads to an expensive (quadratically complex) calculation.

## Distance measurement

Several dimensions can be applied in quantifying similarity between business process models (*Schoknecht et al., 2017*). They focus on different business process model aspects in similarity calculation. These aspects mainly cover the following categories: natural language aspect which can cover syntactic and semantic analysis; graph structure aspect, model behavior aspect, human estimation aspect which may involve expert opinions on similarity, and other aspects. Model clone detection tools may cover one or more of these aspects in calculating model similarities. Our approach mainly relies on the natural language and graph structure dimensions.

Originally, SAMOS applies a distance measurement over the built VSM. Different distance measures such as cosine and Manhattan (*Ladd, 2020*) are available in the framework. For clone detection, the framework offers an extended distance measurement, which we also used in our experiments. The following are some arguments on the distance measurement of the framework:

- For clone detection, a normalized and size-sensitive measure is preferred. There are several available techniques in the literature that meet these criteria. The Bray–Curtis is the one with an R implementation that is used in the framework (*Deza & Deza, 2009*).
- In basic VSM approach, orthogonality is considered, which is about taking into account all columns for distance calculation. In the case of clone detection, this is violated by the

framework, as there is usually (partial) similarity among many model features which would result in unreasonably high similarities between models. Consequently, the union of only the set of features for the two models to be compared are considered, rather than the whole feature set in the dataset.

For these reasons, in previous work, a masked variant of the Bray–Curtis distance (Eq. (5)) was integrated into SAMOS, which extends the distance function available in the R package *vegan*. Given an $N$ dimensional vector space, with $P$ and $Q$ as data points for BPMN models representing the whole model, or smaller scopes such as lane or subprocess; $P$ consisting of features $P_1,\ldots, P_m$ and $Q$ consisting of features $Q_1, \ldots, Q_n$, $p$ and $q$ the corresponding vectors on the whole vector space for $P$ and $Q$, the masked Bray-Curtis distance is measured on the vector subspace $P \cup Q$ (size $\leq m + n$) as:

$$bray'(P, Q) = \frac{\sum_i^{P \cup Q} |p_i - q_i|}{\sum_i^{P \cup Q} (p_i + q_i)}, \tag{5}$$

## Clustering

As the final step in clone detection workflow in SAMOS, a clustering is done over the calculated distance matrix in the previous step. The goal is to find the groups of data point or model clusters in our case that meet at least a set of requirements. They need to be non-singleton (*size* $\geq 2$) with a minimum size (*size* $\geq n$) group of data points that are similar (*distance* $\geq t$), with n and t thresholds which can vary based on the application scenario. SAMOS already has support for different clustering technique: k-means, hierarchical clustering, and Density-Based Spatial Clustering of Applications with Noise (DBSCAN) (*Ester et al., 1996*). In our experiments, we opt for DBSCAN clustering, as it was also applied in the original study of clone detection in SAMOS (*Babur, Cleophas & van den Brand, 2019*). Also, it seemed to be better suited for clone detection because of the following reasons:

- detecting clusters in various shapes (non-spherical, non-convex),
- detection of noise, *i.e.*, non-clones
- suitability for larger datasets.

## EXPERIMENTAL EVALUATION

In this section, we present three experiments that we perform to evaluate the capabilities of our tool SAMOS and compare it with the Apromore toolset. All the output in the following experiments are produced based on the implemented techniques that are presented in this study. The experiments mainly evaluate the accuracy of the tools in question. The experiments are developed based on the following empirical research questions:

**RQ1:** *How accurately does each tool compare a pair of models?*
**RQ2:** *How accurately does each tool discover clone pairs in a real model dataset?*

**RQ3:** *How accurately can SAMOS detect clones in lane and subprocess decomposition in BPMN models?*

For doing the experiments, we use the extended version of SAMOS along with the approximate clone detection tool and the similarity tool of Apromore.

- **Experiment 1: Mutation Analysis**: In this experiment, we perform an analysis based on an artificially generated mutation model set we ourselves created. During this study, pairwise distances between a base model and its mutants are measured to test the accuracy of our technique in different settings against that of the other tools. This experiment addresses RQ1.
- **Experiment 2: SESE Fragment Clone detection on EPC Models**: In our second experiment, We perform an evaluation on the clone detection accuracy of our tool compared to Apromore on a real model dataset. This experiment addresses RQ2.
- **Experiment 3: Clone Detection on BPMN Models with Lane and Subprocess Scope**: In the last experiment, the aim is to show the strength of our tool beyond what is offered by Apromore. For that, we work on the decomposition capability of SAMOS on the Lane level. This experiment addresses RQ3.

## Experiment 1: mutation analysis

The first experiment is based on a conceptual framework proposed by *Stephan (2014a)*, *Stephan, Alalfi & Cordy (2014)* to validate our approach. The framework promotes the use of mutation analysis for evaluating model clone detection tools and techniques. In this section, we provide the details of our assumptions, case design, and goals. Finally, we present the results obtained and discuss the pros and cons of each tool.

### Experiment design

As an initial step, we make a simplifying assumption that the scope of this experiment is limited to the control flow elements of BPMN 2.0. The reason for this decision is that Apromore supports only these elements in its current implementation. In this experiment, we first build a model mutation dataset using some commonly used mutation operators (*Stephan, 2014b*). The operators cover different notable changes that are possible such as applying changes on a model element and add/move/remove elements, which can better show the shortcomings of each tool. Some additional cases are also included to show more exceptional model change scenarios, similar to what is done by *Roy (2009)*. Finally, all the tools are run using this list of models and results are collected. We evaluate the accuracy of each tool on the mutation dataset. As listing all possible mutations is not feasible, we give a representative list of such common changes applied on BPMN models in Table 2. A description on the mutation sets is given as follows:

- **Set 1a**: This set consists of mainly trivial atomic changes applied on a model. The change operations include various types of single changes, including adding and removing elements, changing their type, and label changes that rename an element to a different name.

Table 2 Pairwise relative distances (reverse similarity) for SAMOS and Apromore.

| Set | Id | Base model | Mutant model | SAMOS unigram | SAMOS bigram | Apromore clone detection | Apromore sim tool v7.15 |
|---|---|---|---|---|---|---|---|
| la | 1 | baseProcess | atomic-changeEventNameRandom | 0.98 | 0.97 | 0.94 | 0.9 |
| | 2 | baseProcess | atomic-addEvent | 0.96 | 0.96 | 0.941 | 0.88 |
| | 3 | baseProcess | atomic-addTask | 0.96 | 0.95 | 0.941 | 0.88 |
| | 4 | baseProcess | atomic-changeOrderTasks | 1 | 0.93 | 0.91 | 0.95 |
| | 5 | baseProcess | atomic-changeTaskNameRandom | 0.98 | 0.94 | 0.907 | 0.87 |
| | 6 | baseProcess | atomic-changeTypeGateway | 0.99 | 0.95 | 0.85 | 1 |
| | 7 | baseProcess | atomic-removeEvent | 0.96 | 0.97 | 0.967 | 0.77 |
| | 8 | baseProcess | atomic-removeTask | 0.96 | 0.95 | 0.935 | 0.87 |
| | 9 | baseProcess | atomic-changeGatewayNameRandom | 0.99 | 0.97 | 1 | 1 |
| | 10 | baseProcess | atomic-changeSeqFlowNameRandom | 0.98 | 0.99 | 1 | 1 |
| | 11 | baseProcess | atomic-changeElementAttribute | 0.96 | 0.99 | 1 | 1 |
| | 12 | baseProcess | atomic-removeSeqFlow | 0.97 | 0.96 | 0.941 | 0.85 |
| | 13 | baseProcess | atomic-changeTypeTask | 0.99 | 0.98 | 1 | 1 |
| 1b | 14 | baseProcess | atomic-changeTaskNameCosmetic-addWhitespace | 1 | 1 | 0.97 | 0.91 |
| | 15 | baseProcess | atomic-changeTaskNameCosmetic-changeCase | 1 | 1 | 0.7 | 1 |
| | 16 | baseProcess | atomic-changeTaskNameCosmetic-addExtraChar | 1 | 1 | 0.75 | 0.97 |
| | 17 | baseProcess | atomic-changeTaskNameSyn | 0.98 | 0.94 | 0.907 | 0.87 |
| | 18 | baseProcess | atomic-changeEventNameSyn | 0.99 | 0.97 | 0.937 | 0.9 |
| | 19 | baseProcess | atomic-changeGatewayNameSyn | 0.98 | 0.84 | 1 | 1 |
| | 20 | baseProcess | atomic-changeTaskNameTypo | 0.98 | 0.94 | 0.996 | 0.98 |
| 2 | 21 | baseProcess | atomic-moveTaskOtherPlace | 1 | 0.93 | 0.91 | 0.97 |
| | 22 | baseProcess | atomic-swapTasks | 1 | 0.95 | 0.94 | 0.97 |
| | 23 | baseProcess-moveTaskToSimilarContext | baseProcess-moveTaskToSimilarContext | 1 | 1 | 0.94 | 1 |
| 3 | 24 | baseProcess | cum-addMultipleElements | 0.83 | 0.81 | 0.791 | 0.6 |
| | 25 | baseProcess | cum-changeMultipleGatewayTypes | 0.98 | 0.89 | 0.978 | 1 |
| | 26 | baseProcess | cum-moveMultipleElements | 1 | 0.85 | 0.82 | 0.87 |
| | 27 | baseProcess | cum-removeMultipleElements | 0.87 | 0.82 | 0.61 | 0.62 |
| | 28 | baseProcess | cum-renameMultipleElements | 0.92 | 0.72 | 0.64 | 0.46 |
| | 29 | baseProcess | cum-changeMultipleElementsAttr | 0.94 | 0.93 | 1 | 1 |
| 4 | 30 | ex-baseProcess-sameElements-diffStructureDiffNaming | ex-sameElements-DiffStructureDiffNaming | 0.69 | 0.3 | 0.4 | 0.28 |
| | 31 | ex-baseProcess-diffStructureSimilarNames | ex-diffStructureSimilarNames | 0.98 | 0.605 | 0.73 | 0.95 |
| | 32 | ex-baseProcess-multiplePartialSimilarNames | ex-multiplePartialSimilarNames | 0.85 | 0.454 | 0.57 | 0.57 |
| | 33 | ex-baseProcess-multiExitGateway | ex-atomic-multiExitGateway-changeType | 0.99 | 0.908 | 0.64 | 1 |
| | 34 | ex-baseProcess-similarStructureDiffNaming | ex-cumulative-changeAllNames | 0.26 | 0.343 | 0.49 | 0.44 |
| | 35 | ex-baseProcess-compareSmallWithBigModel | ex-compareSmallWithBigModel-diffNames | 0.62 | 0.265 | 0.44 | 0.27 |

(Continued)

| Table 2 (continued) | | | | | | | |
|---|---|---|---|---|---|---|---|
| Set | Id | Base model | Mutant model | SAMOS unigram | SAMOS bigram | Apromore clone detection | Apromore sim tool v7.15 |
| | 36 | ex-firstProcess-oneIncludesAnother | ex-secondProcess-oneIncludesAnother | 0.83 | 0.88 | 0.63 | 0.4 |
| | 37 | ex-baseProcess-compareSmallWithBigModel | ex-compareSmallWithBigModel-sameNames | 0.66 | 0.52 | 0.66 | 0.44 |
| | 38 | ex-firstbase-multipleUseOfSameElementName | ex-secondModel-multipleUseOfSameElementName | 0.44 | 0.07 | 0.26 | 0.2 |

- **Set 1b**: The aim of this set is to evaluate the NLP-related capacity of the tools in the form of subtle element name changes: cosmetic renaming (such as camel *vs.* snake case, lower *vs.* upper case), making minor typos, and replacing a word by its synonym.

- **Set 2**: This set covers some main possible movements in a model. For this set we have swapping, and two types of moving model elements: One is moving a model element to another place with different context, and the second is moving to a similar context. In swapping, an element is swapped with another element at a different point in the model. In first type of moving, an element is moved to an arbitrary place in the model, while in the second type, an element is moved to a place in the model with similar neighbor vertices as the ones in its original place.

- **Set 3**: In this set, a base model is modified to include multiple possible changes applied on the model. The aim of developing this set is to examine the behavior of the tools where such accumulated changes are applied to a model. The targeted change operations include adding, removing, moving, renaming multiple elements at different places in the model. There are also cases for changing type of multiple gateways, and changing some element attributes, *e.g.*, adding loop characteristics or conditions on a Task element.

- **Set 4**: This set is added to show some exceptional cases that may not usually happen in real models, but may give us some clue about functioning of the tools and their reaction to such extreme cases. These changes include the following: keeping the same elements, change model structure and change labels; change the model structure using the same elements; change several element names to similar ones; change the type of a gateway that has multiple gates; change all element names in the model; extend model to a bigger model with mostly different labels; extend a model with other elements which still contains the original model as a fragment; extend model to a bigger model with similar names; use the label from the first model in multiple labels in the second model.

For the manual creation of the mutation sets, we use average-size BPMN models as base models (*Kunze et al., 2011*). It is also important to note that atomic mutation operations do not always lead to an atomic change on the model, *i.e.*, a single change on an element may lead to changes on the surroundings of that element. This depends highly on which element you want to modify and the connections between the element and the rest of elements in the model. For instance, a renaming operator often affects a single element, while removing a gateway with several incoming or outgoing connections has a bigger

impact on the model. In the design of our mutation set, we tried to include some of such possible changes to better show the idea.

### Goals for distance measures

It turns out there is no standard way to precisely measure the similarity of a pair of models. However, we can specify a set of distance metrics to approximate the distance between a pair of models or fragments. We are expecting fine-tuned distances for small changes. This is because we want to be able to detect clones where there are multiple such small changes *e.g.*, cosmetic label changes or replacing several labels with their synonym should result in an accumulated small enough distance between two models in order to still be counted as clones. As a rule of thumb, removing an element should result in a higher distance than changing the same element to a similar type, and similarly the latter should result in a higher distance compared to renaming the same element to a synonym. Further details about these measures are provided in *Babur, Cleophas & van den Brand (2019)*.

### Running tools on the mutation set

As expected, each tool runs under some configurable settings. For SAMOS, we consider the Unigram and Bigram settings. The second tool, Apromore similarity plugin, can be run using either of the Hungarian or Greedy algorithms. We run the tool in Hungarian mode. Apromore also allows the user to adjust some thresholds including model, label, and context similarity thresholds (with default values as 0.6, 0.6, and 0.75 respectively) to parameterize the similarity search process. For example, the label similarity threshold provides the similarity threshold of the different labels of a model. The similarity thresholds provide a filtering on the result set. All the three thresholds are set to zero to let models possibly with similarities under the thresholds appear in the result set. For the third tool, Apromore approximate clone detector, the default settings are applied.

Another consideration is the input model type used by the tools. Apromore has developed a canonical format to capture different business process models in a common representation (*La Rosa et al., 2011*). Models from different business process model notations are initially transformed to this format. Apromore offers a Canonizer service which allows for transforming between different notations through the canonical format. Despite that, there is a limitation in the input format for the Apromore tools. The similarity plugin of Apromore reads models in BPMN only, while approximate clone detection tool reads models in EPC format only. To be consistent in the whole experiment, we build the models in BPMN 2.0 notation and transform them to their EPC counterpart through the Canonizer service of Apromore to be used by its clone detection tool.

### Evaluation of the results

Looking at the distance measurements reported in Table 2, it is noticeable that tools are revealing where they are doing better and where are the shortcomings. We can distinguish categories in the result set from a single tool's perspective. There are points at which a tool is doing fine, and other points where the tool is not showing a good result. To get even more insight, we may look at the points where only a tool is able to show reasonable results, while others are failing. In the following, we will have a closer look at these points.

For the following cases, it seems SAMOS is outperforming Apromore:

- **Naming**: The naming in gateways and sequence flows are ignored in Apromore. It also seems SAMOS is able to better handle minor cosmetic changes such as white space, case change, and addition of extra chars on the label.
- **Attributes**: Element attributes (as defined in metamodel) are not taken into account in Apromore.
- **Type change**: As there is a family of Task types in BPMN 2.0, changing from one type to another can be seen by SAMOS but not by Apromore.

The main reason for most of the failures for Apromore mentioned above seems to come from the previously mentioned fact that Apromore is not considering language-specific information in a process model, as they are treating process models in the form of a more generic structure of a process graph captured in their canonical representation.

On the other hand, there is a case that Apromore is able to detect, which is not captured by SAMOS. It is about the moving of an element to another place with a similar context in the model (with same preceding and succeeding neighbors). This originates from a limitation of using Bigram where only two consecutive nodes are considered as a unit of feature extraction in SAMOS. This can be improved by taking bigger chunks of n-grams as features.

Regarding the distance measures specified in "Goals for Distance Measures", the tools seem to show different behaviour about specific mutation operations, but for the majority of the cases, they all show an acceptable sensitivity regarding the impact size of atomic and cumulative change operations. There are still cases at which the tools do not seem to behave as expected. For instance, Apromore clone detection tool seems to show an excessive impact by cosmetic changes (*e.g.*, for case changes). On the other hand, the Similarity tool seems to show a higher sensitivity for adding or removing elements compared to other tools. Also for SAMOS, it seems it is less sensitive about renaming multiple elements. Note that deciding about the right impact size for a modification is not always straightforward. For instance, in the example of atomic changing a gateway type with multiple in or out flow connections, Apromore seems to show much higher sensitivity than SAMOS, but deciding which one is better may be a preference to the user or the case applied.

## Experiment 2: model clone detection on asset management dataset

After obtaining results of the first experiment, we now have a basic idea about the two tools. To further understand the differences between SAMOS and Apromore, we proceed to the second experiment, where we compare the clone detection capability of the two tools by running them on a real dataset. For this experiment, we run the tools on EPC models. The decision is made as Apromore approximate clone detection tool reads only EPC models. For the dataset we do not have many options as EPC models seem to be rarely used nowadays. For this experiment, we chose the Asset Management dataset from the Model Matching Contest 2015 which is derived from the SAP Reference Model Collection

(*Enterprise Modelling and Information Systems Architectures, 2015*). The Asset Management dataset consists of 36 pairs of similar EPC models, resulting in a total of 72 models.

### Methodology

Given the 36 model pairs, we performed the following steps to run the tools on this model set.

1. Apromore clone detection: run clone detection of Apromore with default settings on the models.
2. Fragmentation: extract all SESE fragments from all models using Apromore toolset in the form of separate EPC models.
3. Transformation: transform the EPC models to BPMN models through the Apromore canonical format.
4. SAMOS clone detection: run clone detection of SAMOS with default settings on the BPMN models obtained from the previous step.
5. Validation: inspect random subsets from the three sets obtained from the clustering result of the two tools:

- Clone pairs common for both: SAMOS ∩ Apromore
- Clone pairs only in SAMOS: SAMOS – Apromore
- Clone pairs only in Apromore: Apromore – SAMOS.

Note that the Apromore tool works in the scope of SESE fragments for clone detection, while currently there is no support for this scoping for BPMN models in SAMOS. Therefore, for a comparison between the two tools, we need to obtain the SESE fragments of given models and run clone detection on them. This way, both tools would be running clone detection in the same scoping which makes them comparable. We obtain the SESE fragments identified by Apromore as separate EPC models, which then are transformed to BPMN models to be used in SAMOS (step 2 and 3 in the methodology).

Regarding step 4 (SAMOS clone detection), a pre-processing is done on the distance matrix before applying the clustering algorithm. The aim of this pre-processing is to avoid overlapping clone pairs where one fragment contains another. A definition that considers two fragments as clones only because one contains another leads to many false-positives and should be avoided. This is a known issue in the code and model detection domain (*Pham et al., 2009*). Apromore avoids this problem in its definition and by the techniques such as RPST that they apply. Since in SAMOS we do not support the SESE decomposition, in this experiment, we obtain the fragment containment information from Apromore and use it to mask out overlapping pairs in the distance matrix before continuing with clustering for clone detection.

### Results

The number of generated pairs based on the obtained results are given in Table 3. Considering the one big cluster (with over 500 data points) produced in SAMOS, the

**Table 3  Number of clone pairs found in the Asset Management dataset.**

|  | Type B, C #pairs |
| --- | --- |
| SAMOS | 166,659 |
| Apromore | 898 |
| SAMOS–Apromore | 165,914 |
| Apromore–SAMOS | 153 |
| SAMOS ∩ Apromore | 745 |

resulting number of pairs is much bigger in comparison to the one for Apromore. For SAMOS, there are 49 clusters in total, while for Apromore, this number is 156 which is much larger than that of SAMOS. The number of data points in clusters for Apromore is also much lower than that of SAMOS which means the corresponding group of data points were distinctive enough in the default adjusted distance threshold of 0.3.

As mentioned in "Methodology", we relied on a manual validation for measuring the accuracy of the tools. Random subset of common and different pairs from the two tools were inspected. The random sampling was done with a confidence level of 90% and with a 10% margin of error to build the manual validation sets. Table 4 shows the resulting set sizes. Two of the models were incompatible with Apromore, thus being removed.

A preprocessing was done to remove possible duplicates from the original sets. Also, before doing the sampling in the produced three sets, the exact clone (Type-A) pairs were excluded from the lists. To do that, originally an internal function of the Apromore tool was used to get the list of exact clones. However, by checking the database of the tool, we noticed the existence of additional fragment pairs with distance 0 that were missing in the original results, which were also added to the list.

We use stratified sampling in this experiment, meaning the whole clone pair set is treated as subgroups which correspond to clusters and sampled separately for each cluster. This way the samples represent best the whole population set.

Apromore by default allows for overlapping in clusters, which may result in repetition of clone pairs. With a separate filtering pass, the duplicate pairs are removed.

For a quantitative evaluation of the results we use precision and recall measures as well as the F-score (harmonic mean of the recall and precision) (*Larsen & Aone, 1999*). Precision and recall are widely used performance metrics in Information Retrieval and Machine Learning that apply to data retrieved from a collection. Precision is the fraction of relevant instances among the retrieved instances, while recall is the fraction of relevant instances that were retrieved. Relevant instances correspond to real clones in our case. Table 5 shows the aggregated results from all the validation sets for precision, relative recall and the F-score separately for SAMOS and Apromore. The calculation of the precision, relative recall, and F-score are based on the sample sets shown in Table 4. In this experiment, we follow a similar approach for calculating the mentioned measures as in *Babur, Cleophas & van den Brand (2019)*.

**Table 4 Size of the validation sets.**

|  | Type B, C |
|---|---|
| SAMOS–Apromore | 69 |
| Apromore–SAMOS | 48 |
| SAMOS ∩ Apromore | 63 |

The calculation of the precision for each tool is based on the samples taken from the relevant sets for the tool. For instance, in the case of SAMOS, we look into the two sets "SAMOS–Apromore" and "SAMOS ∩ Apromore" sets. Regarding the recall, while we cannot assess the absolute recall, we give an assessment of (relative) recall in this experiment. The number of real clones (as labelled manually) required for the relative recall is obtained through the following steps: (1) obtain the percentage of the relevant clones in all the validation sets, (2) extrapolate the percentages to the original sets regarding the validation sets (*i.e.* multiply percentages with the sizes) and (3) take the average over all the sets. Regarding the extrapolation step, we rely on the underlying assumption that choosing a statistically significant sample size for valiation sets, well represents the population, *i.e.*, the full set of clones, from which the samples were taken.

The calculation of the mentioned measures for each tool is based on the formulas given in Eqs. (6) to (10). Given the three sets defined before, SAMOS–Apromore (set1), SAMOS ∩ Apromore (set2), Apromore–SAMOS (set3), the letters used in the below equations mean the following: $T_i$ corresponds to the number of true positives (correctly identified as clones) in the i$^{\text{th}}$ set, $N_i$ corresponds to the total number of elements in the i$^{\text{th}}$ set, and $S_i$ corresponds to the number of elements in the sample set of the i$^{\text{th}}$ set.

$$SAMOS\_precision = \frac{(T_1 * N_1)/S_1 + (T_2 * N_2)/S_2}{N_1 + N_2} \tag{6}$$

$$SAMOS\_relRecall = \frac{(T_1 * N_1)/S_1 + (T_2 * N_2)/S_2}{(T_1 * N_1)/S_1 + (T_2 * N_2)/S_2 + (T_3 * N_3)/S_3} \tag{7}$$

$$Apromore\_precision = \frac{(T_2 * N_2)/S_2 + (T_3 * N_3)/S_3}{N_2 + N_3} \tag{8}$$

$$Apromore\_relRecall = \frac{(T_2 * N_2)/S_2 + (T_3 * N_3)/S_3}{(T_1 * N_1)/S_1 + (T_2 * N_2)/S_2 + (T_3 * N_3)/S_3} \tag{9}$$

$$F - Score = \frac{2 * precision * recall}{precision + recall} \tag{10}$$

Note that the precision and recall in Eq. (10) are the metrics calculated separately for each tool as reported in Table 5. For intacne, in the case of SAMOS, the precision in the formula corresponds to SAMOS_precision, and the recall corresponds to SAMOS_relRecall, calculated using Eqs. (6) and (7) respectively.

**Table 5 Aggregated results from all the validation sets for precision, relative recall, and F-score of SAMOS and Apromore.**

|          |            | Type II, III |
| -------- | ---------- | ------------ |
| SAMOS    | Precision  | 0.638        |
|          | Rel. recall | 0.998       |
|          | F-Score    | 0.778        |
| Apromore | Precision  | 0.920        |
|          | Rel. recall | 0.007       |
|          | F-Score    | 0.014        |

### Discussion

In this experiment, a comparative clone detection was performed on a real dataset. Overall, Apromore outperforms SAMOS in precision, but the results show a much higher relative recall for SAMOS.

Given the rather small dataset used in this experiment, the exact runtime of the tools is not provided. Both tools get slower as the input size increases. However, for this experiment, both tools ran in order of few minutes. For SAMOS, the VSM computation and the distance calculation are the main parts that slow down the tool. The complexity of the tool is quadratic with respect to the total feature set size of the model dataset. The feature comparison techniques such as Hungarian algorithm and the applied NLP techniques also increase the computation time. While for Apromore, it seems one of the main bottlenecks is about the costs related to the incremental building of the RPSDAG (*Dumas et al., 2013*). We plan to improve our tool performance by applying optimizations on the used techniques and by a distributed computing.

One of the causes for the lower precision of our tool in this experiment seems to be originating from the fact that our current implementation performs best when the majority of the model elements are labeled. EPC models do not support labels for connectors (gateways in BPMN) and sequence flows; since we used EPC models for this experiment, this seems to also have a negative effect on the final results. In the next experiment, we use BPMN models with different fragmentation techniques, where we get a better precision compared to this experiment.

### Experiment 3: clone detection with lane and subprocess scoping

We perform another experiment to demonstrate the potential of our tool in clone detection for different model scoping settings. Basically, scoping in SAMOS boils down to how granular we want the feature extraction to be. For instance, this can be for the whole model which is assumed to be the default mode, or it can be for smaller fragments in the model. It all depends on how we want to slice a given model into logically connected elements for similarity detection.

In the case of BPMN models, we can think of different ways of decomposing models into smaller parts. For this experiment, we look into Lane and Subprocess elements in BPMN specification. They are specifically chosen as both of them provide a form of logical

grouping with sub-elements. In this experiment, we limit the results of clone detection for the mentioned decomposition styles, but the idea can easily be generalized for other ways of fragmenting process models.

### Methodology

In the following, we give an outline of the followed steps for this experiment. The steps are repeated for both of the decomposition styles we chose for this experiment.

1. GitHub search: search for models containing the decomposition element in the model.
2. Model filtering: filter out all invalid, irrelevant, too small, non-English, and duplicate models.
3. Clone detection: run the tool with the default settings on the model set to find type-A (exact clone), and type-B and type-C (approximate clones), among the model fragments. The Bigram encoding with a distance threshold of 0.30 is applied.
4. Validation: manually inspect a randomly selected sample set of the clone pairs (with confidence of 90% and 10% margin of error) to find true/false clone pairs.

Note that for this experiment, similar to the previous experiment, we address the overlapping clone issue. This means that, given a pair of lanes, if one lane contained the other, the pair is considered as overlapping clones, and thus it is excluded from the result set. Similarly, this is done for Subprocess clone pairs.

### Model collection

For this experiment, we needed to run our tool on two sets of models separately for each decomposition setting. As we wanted to run the experiment on real models, we decided to collect models mainly from GitHub. The model collection was done in October 2021. For each set, all models needed to have the corresponding decomposition element, *i.e.*, for Lane decomposition, all models needed to have at least one lane, and the same for the Subprocess decomposition. For that, we did separate GitHub searches: To get models with a *Lane* element, we used the GitHub advanced search with the keyword "bpmn:lane", with *bpmn* file extension, and different size settings to collect a more diverse model set: 0 to 100 kb with 10 kb intervals. Similarly, to get models with a *Subprocess* element, we repeated the same search process with the keyword "bpmn:subprocess". In each search, the candidate models were collected randomly from separate result pages. For Lane decomposition, 395 models were collected in total. After a filtering, 171 models remained for the clone detection process. In the case of Subprocess decomposition, 270 models were collected in total, from which 28 models remained after the filtering process.

To get a list of suitable models for clone detection, a filtering is applied on the collected models. We use the following criteria for filtering:

- Remove models not in the BPMN 2.0 XML serialization format,
- Remove models that are too small,
- Remove models with non-English labels,
- Remove duplicate models.

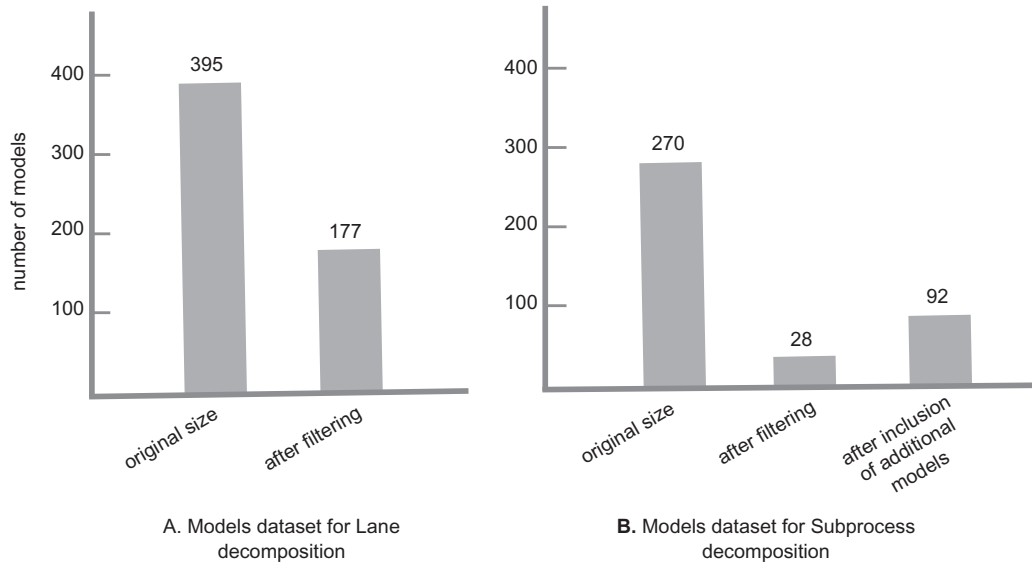

**Figure 5 Model datasets for lane and subprocess decomposition.**

For the Subprocess case, as the final list of models after the filtering was small, other valid models from the Lane search with Subprocess element, as well as an additional set of models that were collected from the RePROSitory open access business process model database (*PROS-Lab, 2019*) were added to the list. The final list totaled 92 models. As a final note, during this experiment we do not check the models for any possible syntactical or semantical rule violations, as this is not a major issue for our clone detection process. Figure 5 gives an overview of the model sets for each decomposition.

### Clone detection with lane decomposition

Lanes in BPMN correspond to roles in organizations. Lanes consist of a set of activities that are handled by an actor, such as a person or a system. They are represented by a box-shaped graphical element with a clear boundary around it. Lanes seem to be a good candidate to decompose process models, as typically there are roles in business process models that take care of a set of actions. The rationale behind the lane decomposition for clone detection is to find out about other roles who take care of similar sets of activities. This, for example, allows us to further analyze the connections between such roles or to find out about alternatives for a specific role in a business process.

*Results*

Starting with 172 models and following the clone detection steps, the clustering resulted in 59 clusters. Figure 6 shows an overview of the clustering result, labeled with the number of data points (*i.e.*, each as a lane fragment) identified in each cluster. The cluster content were then all converted as pairs. This resulted in a total of 7,590 pairs. Based on a 90% of confidence threshold and a 10% error rate, 68 sample pairs were randomly selected from the whole pair list. With 52 pairs manually evaluated as True clone pairs, the resultant precision was about 76%.

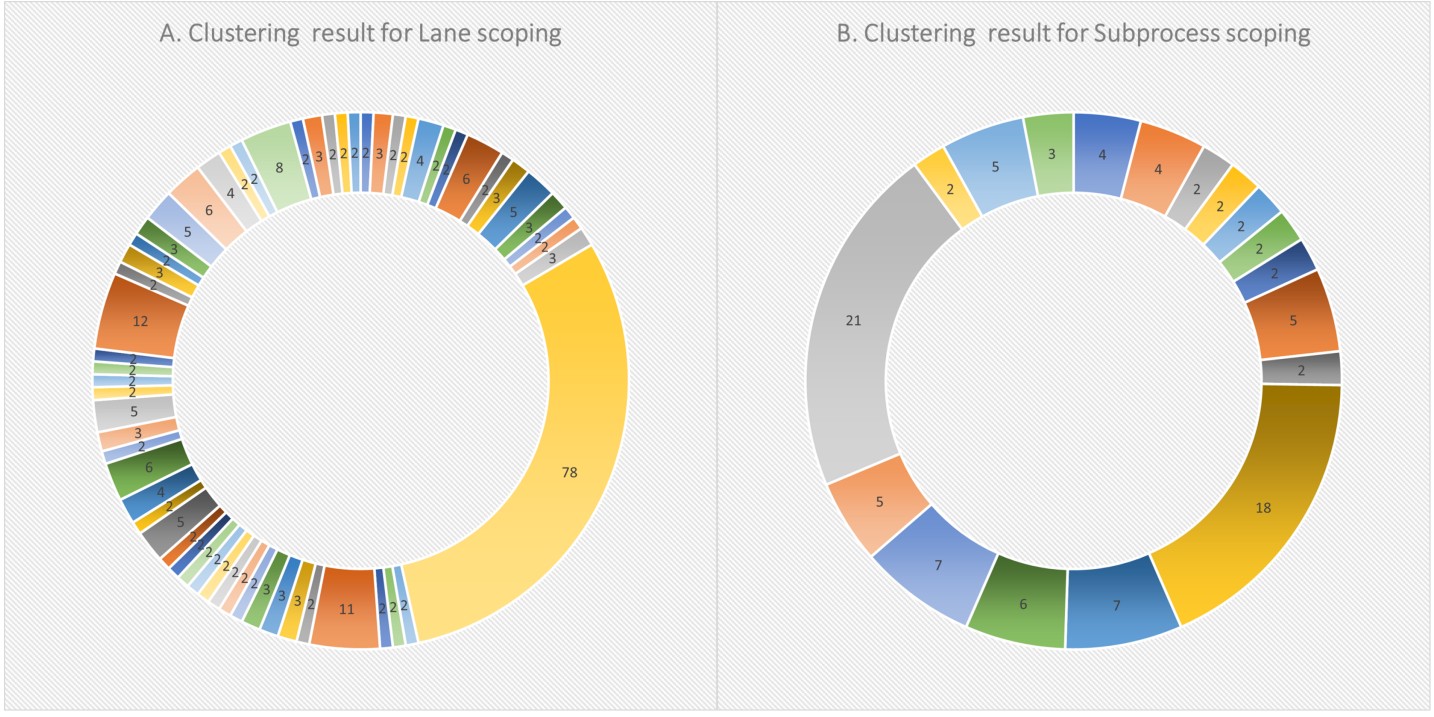

**Figure 6 Clustering results for (A) lane scoping, and (B) subprocess scoping in SAMOS.**

### Clone detection with subprocess decomposition

We repeat the experiment for Subprocesses, similar to what we did for Lanes. Subprocess is another self-contained element in BPMN that are used to simplify the development and use of business process models. They resemble functions in programming languages with an input and output. Subprocess seems also to be another reasonable candidate for model decomposition.

*Results*

In this case, we had collected 92 models in the model set. The clustering phase resulted in 18 clusters, which then again were converted as pairs. Figure 6 shows an overview of the clustering result, labeled with the number of data points (*i.e.*, each as a subprocess fragment) identified in each cluster. This resulted in a total of 472 pairs. Based on a 90% confidence threshold and a 10% error rate, 60 sample pairs were randomly chosen from the whole pair list. With 52 pairs manually evaluated as True clone pairs, the resulting precision was about 87%.

### Discussion

Overall, our tool shows an acceptable precision for both of the scoping settings. Obviously, there is much improvement in the precision from the 2nd experimental evaluation. As noted in the 2nd experimental evaluation, our clone detection tool shows better results when the majority of model elements are labeled, which leads to a better distinction

between model elements in the process of clone detection. This seems to be the reason to get better precision for this experiment. As shown in Fig. 6, there are two broader clusters (encompassing more data points) in each scoping. This is partly due to the use of DBSCAN clustering algorithm, as it tends to include a chain of data points in a single cluster, where the two ends may not be highly similar compared to the neighboring data points. Evaluating alternative clustering techniques, such as hierarchical clustering would be an interesting future work to find the best possible technique for this problem.

A design decision for the decomposition that has a significant impact on the tool accuracy is how we want to include N-grams on the boundary of the selected scope. This happens when there are connections between elements inside and outside a given scope. This is especially the case for the lane decomposition, where there are typically multiple incoming and outgoing connections to/from a lane. For the lane decomposition, our current design allows Bigrams in the extracted features where, *e.g.*, the first part of the Bigram is inside a lane, and the second part is outside it, with a connection between them. This design decision seems to also have negative implications on the tool accuracy. As an alternative, the design could be changed so that such features on the boundary would be excluded.

We use stratified sampling in this experiment, which means the whole clone pairs coming from different clusters are not treated as a single list but as subgroups, thus the sampling is done separately for each cluster. This way, the sample set gives a better representation of the whole population set.

As there is no BPMN clone detection tool with such a scoping capability, we are not able to measure a relative recall similar to what was done in the 2$^{nd}$ experimental evaluation.

## OVERALL DISCUSSION

In this section, we discuss some important aspects of our approach and the developed tool.

### Underlying framework

The presented technique in this article is built on top of SAMOS, which enables us to exploit its capabilities in NLP and statistical algorithms. The extensibility of the framework allows us to add feature extraction for BPMN specification and customize and add new distance measures. With the support of R in the back-end, we are able to do more advanced statistical and data mining techniques. We've been working on integrating our tool along with other related tools under the Eclipse Arrowhead framework. The other tools function as model repository, model management, and visualizing dashboards that interoperate as a toolchain through the Eclipse Arrowhead framework (*Arrowhead, 2021*).

### Accuracy

Our approach as implemented, shows an acceptable accuracy as reported under the experimental evaluations. Apromore clone detection tool shows a better precision compared to our tool, however, regarding the recall measure, our tool yields better results. Our assessment on the recall is relative to the tool results which is reported in the 2$^{nd}$

experimental evaluation. A qualitative analysis on the cases where both the tools show weaknesses will help us to work towards a better performance in our tool.

## Performance and scalability

Overall, performance for both SAMOS and Apromore can be improved. For SAMOS, this is mainly related to the quadratic complexity of VSM and distance calculations, employing a complex NLP, and the comparison algorithm performed. For Apromore, there are partly similar problems as calculating different levels of distance measurements and seemingly the most expensive task of building the RPST and related calculations on top of that. Persisting and retrieving all the data into/from a relational database during the clone detection process may also have an extra added burden to the time complexity of the algorithm. SAMOS has already been tested for its scalability to handle quite large datasets (in the order of tens of thousands of models), although with optimizations and in iterative mode (*Babur, Cleophas & van den Brand, 2019*). For Apromore, as claimed in *La Rosa et al. (2015)*, the RPSDAG implementation employed by the tool can handle repository sizes in the magnitude of hundreds of models. The authors intend to improve on the scalability with some optimizations regarding the distance calculation.

## Genericness

SAMOS, the underlying framework, is in principle generic, meaning it can be used for any graph-based modeling notation. As the starting point for the analytics in the framework is based on the extracted features from models in an encoding defined in the framework, in practice, it is enough for one to generate their so-called features, and use the rest of the framework as-is without the need to modify it in a domain-specific way. As stated in the article, our feature extraction relies on and covers main parts of the BPMN 2.0 metamodel for process modeling.

## Clustering

An important aspect of our approach is about clustering. Handling of the connectivity in the used clustering technique as spherical/convex *vs.* non-spherical/convex shapes influences our clustering results. This is a known issue in the data mining domain, but not much studied in the clone detection domain. As it appears in the $2^{nd}$ experimental evaluation, among our clustering results we encountered a single very big cluster compared to other clusters, which seems to be a result of the same issue. We plan to tackle this issue more by comparing our clustering technique with others such as Hierarchical clustering techniques.

## Improving on SAMOS

A set of custom-tailored improvements are added to SAMOS to serve better our business process model clone detection. These changes and additions are specifically done to improve the distance measurement calculations. After inspecting the results from the experimental evaluations, we further plan to improve on other aspects of the framework. More work can be done regarding the use of NLP for more accurate results on label similarity measurements. For their semantic similarity measurement, inclusion of domain-

specific ontologies could also be considered. Another improvement could be about optimizations in the VSM construction to lower its time and space complexity as this is the most costly step in the clone detection workflow of SAMOS.

### More powerful NLP

Having a strong NLP component in place is central to a quality clone detection tool. Obviously, labeled elements are frequently used in business process models and play an important role when comparing model elements. SAMOS already utilizes some commonly used NLP functionality such as tokenization, stemming, lemmatization, and Wordned-based measures, but there is still room to strengthen this by adding more advanced features such as context-based or domain-specific semantic checking.

### Other practical aspects

There are various aspects to consider when applying model clone detection in practice (*Deissenboeck et al., 2010*; *Stephan, 2014a*). One of the known issues is that of overlapping clones which is common in the code and model clone detection domain. In our experimental evaluation, we try to partially address this issue, but still more work needs to be done in this regard. Also, we have ongoing work such as visualizing clone detection results for a more convenient inspection of the results.

### Threats to validity

A threat to validity of this work is regarding the usage of the Apromore tool in our experiments. The clone detection tool developed by the Apromore constitutes a part of the much bigger ecosystem under the Apromore platform. Although there are multiple publications on the approach and the utilized techniques, still there is no documentation on different aspects of the tool in implementation level and how to properly run the tool for clone detection as intended. We ran the tool with the default settings for clone detection and obtained the results accordingly, as reported in the experimental evaluations.

Another threat to validity of this work is the absence of measuring the absolute recall, although we compared our technique to the state-of-the-art Apromore clone detection tool. There is no other tool against which we can compare our tool, but for a better assessment of our tool, we can follow a more automated mutation analysis (*Stephan & Cordy, 2019*). We also applied a manual validation approach to assess the precision of the tool, which is an error-prone process as it is mainly a labor-intensive activity. To overcome this, we plan to have multiple assessors do the validation, also get the help of domain experts from the community and the industry, to build more precise knowledge about the notion of BPM clones (*Stol, Ralph & Fitzgerald, 2016*).

## RELATED WORK

There has been a wide range of research activities going on about clone detection. We will refer to some related work in this domain. The works in clone detection mainly stem from the bigger software clone detection (*Koschke, 2007*) field, and the majority of the work done in the literature is based on code clone detection (*Roy, 2009*). Clone detection in software repositories has also been an active research filed over the last years (*Koschke,*

*2008*; *Roy, 2009*; *Rattan, Bhatia & Singh, 2013*). In the broader view, the research in model clone detection can be considered as a subdomain of model comparison (*Stephan & Cordy, 2013*). Also, our study is closely related to the similarity search field (*Shimomura et al., 2021*). There are similarities in the applied techniques, as for example, how complex data are represented in terms of feature vectors for comparison. In clone detection, we also look for similarities between models, however, in our case, there is no such a query model to search among a set of models. Our approach aims to detect similar models and group them as separate clusters. We look at some relevant studies applying different code and model clone detection techniques.

There are numerous approaches for code clone detection, as reported in *Baxter et al. (1998)*, *Jiang et al. (2007)*, *Gabel, Jiang & Su (2008)*, *Krinke (2001)*. Some of them are using tree-based techniques. For instance, *Baxter et al. (1998)* propose a clone detection approach based on abstract syntax tree (AST) for detecting exact and near-miss clones for arbitrary fragments of program source code. In the process, source code is parsed and an AST is generated, which then is used to find similar subtrees on it. Another tree-based approach is DECKARD (*Jiang et al., 2007*) which detects code clones again using ASTs in which subtrees are characterized as numerical vectors in the Euclidean space and using algorithms to cluster them based on their Euclidean distance.

There are other works for code clone detection using graph-based approaches. In *Gabel, Jiang & Su (2008)*, they offer an extension of DECKARD to deal with program dependence graphs (PDGs). In the process, they first extract a group of subgraphs with potential clone candidates. Then, a group of ASTs are generated from the selected subgraphs. After that, the approach in DECKARD is applied to identify clones. The approach is based on the semantics of PDGs and is not directly applicable to process models. The proposed technique is scalable to hundreds of thousands of models, which is similar to SAMOS in that case. Another approach for clone detection in PDGs is the work by *Krinke (2001)* which is a heuristic based approach for finding similar subgraphs in PDGs, therefore it considers not only the syntactic structure of programs but the semantics (using data flow) is also captured. The approach aims to find maximal isomorphic graphs in fine-grained program dependence graphs.

There are also clone detection approaches in MDE. In *Deissenboeck et al. (2008)* the authors present CloneDetective, which is a method for clone detection of Matlab/Simulink/TargetLink models widely used in the automotive domain. The proposed approach is claimed to be applicable to most data-flow graph-based languages. They partition models into connected components, which are compared using a heuristic matching algorithm. A clustering algorithm is used to group the components based on the sets of their node labels. As claimed in *Pham et al. (2009)*, CloneDetective has a tendency to detect as large clones as possible, which means smaller clone pairs are absorbed by the bigger ones. In our approach, we produce clone pairs for different model granularities to avoid this issue. The work in *Pham et al. (2009)* introduces a tool named ModelCD with two techniques namely eScan and aScan for exact and approximate detection of clones in Matlab/Simulink models. Our work is similar to this work in the sense that they also represent graphs by a set of vectors built from graph features, *e.g.*, vertex in/out degrees and

path lengths. Another work done in *Störrle (2013)* applies to UML domain models such as class or activity diagrams. In this work, they form fragments using objects, their properties, and child objects and measure similarity of fragment pairs. The fragment similarity is calculated through adding up pairwise similarities of their elements. Unlike our approach in SAMOS, they do not take into account the structural similarity and there is not clustering technique applied. SIMONE uses another approach for finding near-miss clones, which is presented in *Alalfi et al. (2012)*. The work provides an adaptation to the code clone detection tool NICAD to detect structurally meaningful near-miss clones in graphical models. This is done by transforming the graph-based models to a text form and using NICAD afterwards. The technique is applied for Simulink models as Simulink already provides a textual representation, but could be extended for process models if a normalized textual representation of such models were generated. From this aspect, this is similar to SAMOS where also models are transformed into a set of textual features for clone detection. Finally, in *Dijkman et al. (2011b)* they identify refactoring opportunities by suggesting pairs of similar process fragments to the user. Unlike our work in SAMOS that considers labels, types and structure in models, they look exclusively at labels. Again, they're not also doing clustering, but rather a pairwise fragment comparison.

Looking at some mentioned approaches, they may have the potential of being extended for applying to business process models, currently there is no such a tool-based approach except the existing one offered by Apromore. However, the approach in Apromore still treats process models in a generic way, *i.e.*, considering their universal internal representation for process models, which limits the tool applicability when it comes to specific languages such as BPMN. In our approach however, although SAMOS, the underlying framework, is generic, our extension is completely geared towards BPMN language, which makes it a language-specific clone detection approach.

## CONCLUSION AND FUTURE WORK

In this article, we present an approach for conducting clone detection on BPMN models. Our approach relies on the techniques implemented in SAMOS. SAMOS is a generic framework for large-scale analysis of models based on information retrieval and machine learning techniques. We present the underlying techniques utilized in the framework, and show how we extend it for clone detection of BPMN models. We have extended SAMOS with customized feature extraction, comparison schemes, customized distance measures and additional scoping in the context of model clone detection.

We evaluate our approach along with the starte-of-the-art business process analytics platform, Apromore, through three different experiments: first one, using a set of synthetic mutation set; second one, using a real model dataset; and third one, using collected models from the GitHub. In the first experiment, we do a comparative evaluation of the tools based on their accuracy in pairwise model similarity. As the results indicate, SAMOS shows a better performance regarding the coverage of the metrics in pairwise model similarity. The second and third experiment aim to provide an evaluation on the clone detection approaches taken by the tools. In the second experiment, we use a dataset from the known SAP Reference Model collection. EPC Models are fragmented as SESE fragments for this

experiment, as this is the only scoping supported by Apromore, the other tool involved in this study. In this case, the results reveal a better precision for Apromore, while a higher recall measure for SAMOS. Our third experiment covers other possible BPMN scoping evaluations in clone detection (*i.e.*, lane and subprocess scoping styles) which covers additional capabilities of our approach. For the last case, we achieve more promising results compared to the second experiment. The results, as well discussions on the tool performance, are provided for each experiment.

There are several possibilities for future improvements. One of them is about scoping. In the current implementation, our tool supports setting of a fixed scoping at each run. However, it would be useful to have a dynamic scoping in place where the tool would automatically adjust the possible scoping settings or find the best scoping and run accordingly.

A second improvement is to consider extracting model features with bigger structures. The results of the current experimental evaluations show that feature extraction in Unigram and Bigram levels do not seem to suffice for getting the best possible results for similarity measure. The next step would be to try with more complex features, such as trigrams or subtrees.

Another interesting future work would be to include execution semantics in clone detection. In that case, the clone detection tool would be able to detect process models that are similar in functionality, but not necessarily with a lexical and syntactical similarity, meaning that they can have different number and type of elements or structurally different. Considering the wide variation in purposes and practices for business process model development, it becomes challenging to detect semantic clones in such models.

Another future improvement can be possible optimizations on the VSM building and application, *e.g.*, through dimensional reduction or handling sparse matrices. As VSM building and the calculations thereof play a central role in our approach, and considering the fact that increase in VSM size leads to more time and space complexities, any optimization on it would significantly reduce our costs.

### Funding

This research is funded by ECSEL, the Electronic Components and Systems for European Leadership Joint Undertaking under grant agreement No 826452 (Arrowhead Tools project), supported by the European Union Horizon 2020 research and innovation programme and by the member states. The funders had no role in study design, data collection and analysis, decision to publish, or preparation of the manuscript.

### Grant Disclosures

The following grant information was disclosed by the authors:
ECSEL.
Electronic Components and Systems for European Leadership Joint Undertaking: 826452.
European Union Horizon 2020 research and innovation programme.

## Competing Interests

The authors declare that they have no competing interests.

## Author Contributions

- Mahdi Saeedi Nikoo conceived and designed the experiments, performed the experiments, analyzed the data, performed the computation work, prepared figures and/or tables, authored or reviewed drafts of the article, and approved the final draft.
- Önder Babur conceived and designed the experiments, analyzed the data, performed the computation work, prepared figures and/or tables, authored or reviewed drafts of the article, and approved the final draft.
- Mark van den Brand conceived and designed the experiments, analyzed the data, authored or reviewed drafts of the article, and approved the final draft.

## Data Availability

Data and code are available at Zenodo:

Saeedi Nikoo, Mahdi, Babur, Önder, & van den Brand, Mark. (2022). Clone Detection for Business Process Models - supplemental material [Data set]. Zenodo. DOI 10.5281/zenodo.6630475.

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
