# Peer review of "Clone detection for business process models"

_PeerJ Computer Science, doi:10.7717/peerj-cs.1046_

## Round 0.1 · original submission · Major Revisions

Detailed comments have been received for the manuscript. Please prepare a revision and a detailed response letter. Thanks.

Reviewer 1 ·

Basic reporting

In this paper, the authors extended their existing work, a framework named SAMOS (Statistical Analysis of Models) for clone detection of business process models. Originally, the framework was developed to support different types of analytics on models. Though the paper addresses an important issue in MDE, it lacks maturity and merit. Among others, I can see that there are many technical obscurities, which hamper the readability. Moreover, there are various issues with the writing. Altogether, this makes the paper difficult to understand.

Experimental design

- The formulas from (6) to (9) are given without any explanation. First they are different from the precision, recall scores, which are widely used in Information Retrieval and Machine Learning. Moreover, it is not clear why there are different definitions for SAMOS and Apromore. In Formular (10), there is the F-score computed with 'precision' and 'recall', however there is no exact definition of the two metrics (Actually there are 'SAMOS precision', 'SAMOS_relRecall', 'Apromore precision', 'Apromore relRecall', but not 'precision' and 'recall').
- In the evaluation, the authors compare SAMOS only with Apromore. I can see that the authors already published a similar approach, named DeepClone with PeerJ in the following paper: "Clone-advisor: recommending code tokens and clone methods with deep learning and information retrieval" (https://peerj.com/articles/cs-737/). The two approaches seem to be very relevant to each other, though they are used for different purposes. I am wondering if DeepClone can be used to detect clones in business process models.

Validity of the findings

- A critical point is I can see that the improvement is incremental. SAMOS was already presented in a previous work of the same authors, and in this paper, the authors just tailored it to detect clone in models. It is then necessary to highlight the main differences between the two versions, otherwise the novelty of the work becomes questionable.

Additional comments

- The paragraph starting from Line 74 gives a summary of the sections presented in the paper. However, the corresponding numbers are missing. For instance, "In section we provide some introduction BPMN." Similarly, other references across the paper also miss the numbers. This happens because there are no numbers used for the sections. Thus, please consider revising the references.
- Line 303: "first one" --> "The first one"
- Line 304: "Second one" --> "The second one"
- Line 435: "compare it against the Apromore toolset" --> "compare it with the Apromore toolset"
- Line 573: "For SAMOS, in total there were 49 clusters, while for Apromore, this number was 156 which is much bigger than SAMOS." --> "For SAMOS, there were/are 49 clusters in total, while for Apromore, this number was/is 156 which was/is much larger than that of SAMOS."
- Line 574: "The number of data points in clusters for Apromore were also much lower than that of SAMOS ..." --> "The number of data points in clusters for Apromore was also much lower than that of SAMOS ..."
- Line 580: "Two of the models were incompatible with Apromore thus removed." --> "Two of the models were incompatible with Apromore, thus being removed."
- Line 600: "Apromore is outperforming SAMOS in precision" Why using the present continuous tense here, why not the present simple tense? --> "Apromore outperforms SAMOS in precision"
- Line 602: "the exact runtime of the tools are not provided." --> "the exact runtime of the tools is not provided."
- Elsewhere: "Github" --> "GitHub"
- As the final remark, I suggest the authors involve a fluent English speaker or editing service in proofreading the manuscript. At its current status, the paper contains several issues related to the English language. For instance, there are mistakes related to subject-verb agreement. Furthermore, the authors use the past tense intensively throughout paper, which somehow makes the reading become 'hard'. Why not using the present tense instead? For instance, the following sentences starting from Line 706: "The presented technique in this paper is built on top of SAMOS, which enabled us to exploit its capabilities in NLP and statistical algorithms. The extensibility of the framework allowed us to add feature extraction for BPMN specification and customize and add new distance measures." --> "The presented technique in this paper is built on top of SAMOS, which enables us to exploit its capabilities in NLP and statistical algorithms. The extensibility of the framework allows us to add feature extraction for BPMN specification and customize and add new distance measures."

·

Basic reporting

In general, the main weak points of the article are the language and structure. Especially in the introduction, there are many typos and grammar errors. I recommend thorough proofreading.

Regarding the structure, the article is rather hard to follow. The main problems are the insufficient statement of the goal at the beginning of the paper and formulation of the context with respect to the existing capabilities of SAMOS. Both are addressed later in the paper, but it makes it harder for a reader to follow. Lastly, the description of decisions and evaluation is very distributed. A claim or issue is often explained in another section.

English
• Inconsistency in naming (e.g., model-driven engineering)
• Typography – Capital letters (e.g., line 38, 524)
• Typos (e.g., line 51)
• The term “homologous” might not be understandable to a reader (line 39)
• Missing commas (e.g., lines 42,45)
• Ambiguous language – You should be more exact in the text (numbers, examples)
o Line 73 - We use synthetic and real-life datasets in our experiments.
o Line 66 - SAMOS is a framework for conducting different types of analytics on models.
o Line 74 - ... we provide some introduction ...
o Lina 256 – There are few key points
• The use of abbreviations in equations seems inconsistent. Furthermore, it is not entirely clear what is meant by the specific similarity metric (e.g., eTypeSim). While reading the rest of the section, the naming is clearer. But still, I would recommend explicitly declaring the names before using them.
o Eq 1.: sameSime, typeSim, eTypeSim which is also present in others
o Naming AseqFlow and seqFlowB on line 334 also seems inconsistent
• I would discourage directly addressing the reader in the text (line 504).

References
• In the introduction, there are a lot of “could” scenarios of what is possible with clone detection. However, there are little-to-none references supporting this claim.
• Line 82-83 Give an example of the scenario.

Structure
• Missing in-article references (last paragraph of Introduction)
• The label on listing 1. does not seem to reflect the content
• In the methodology section, there is both an explanation of the basic SAMOS approach and the modifications specific to BPMN. While both are important to cover, I would recommend their separation.
• Furthermore, the methodology should start with explicit elicitation of the goals to enhance readability.
• In general, the reasoning behind decisions is hard to follow. For example, in line 350, there is a mention of omitting eType metric. But the reasoning is explained in a different section in line 375.
• A similar issue mentions transformation to EPC format (Lines 493-495), which is explained in a later paragraph (lines 514-522).

Self-contained work
• The introduction should be more specific in the scope of the extension. As evident from the other papers on SAMOS framework, clone detection is the principal analysis supported by the framework. What exactly is the difference here? Other than application on BPMN 2.0 models. It is more evident at the end of the paper, but so it should be at the start.

All definitions present
no comment

Experimental design

The experiments presented in the paper are well-aimed. Especially the mutation-based approach is a very good choice. However, there are weak spots in the explanation of the methodology and taken decisions, further described below. Most importantly, however, the release of experimental data makes it possible for future endeavours in this area.
• When discussing the classification of the clones, there are rather contradictory statements. In lines 101-102, you claim, “We will use this classification as defined in SAMOS throughout this paper.”, but then on lines 112-113 “Also, according to this scheme, we do not distinguish between Type-B and Type-C as they both correspond to approximate clones in our definition.”. This definition and relationship co SAMOS classification schema should be explained in more detail.
• I would recommend explicitly stating the scoping for Apromore.
• I like the division into sets, representing different aspects of mutations in experiment 1.
• The choice of validation using mutation is intriguing. However, it should be supported by a more detailed clarification of the used mutations and their extent in the dataset. This could help in understanding the behaviour, strengths, and weaknesses of your approach. The rough idea was followed in Table 2, but more details are needed.
• I would like to have deeper reasoning behind the choice of the particular concepts depicted in Figure 1. While I understand the decision, I think it could be beneficial to have a deeper discussion regarding the inherent limits of the approach.

Originality
• More attention should be put on the distinction of your contribution with respect to the existing SAMOS framework and Apromore. While it is present in the description of the modifications to SAMOS and evaluation of the experiments, it should be explicit at the beginning.

Well-defined RQ
• Line 58 – “This study is a first step towards that goal.” It is not evident what is meant by the goal.
• While the research questions for the experiment are explicitly stated, I am missing the explicit overarching goal of the paper. While it is present in the text, the information is rather distributed.

Rigorous investigation performed
• You claim that there is just one publicly available tool for clone detection. However, there exists another tool described here: “Skouradaki, M., Andrikopoulos, V., Kopp, O., Leymann, F. (2016). RoSE: Reoccurring Structures Detection in BPMN 2.0 Process Model Collections. In: , et al. On the Move to Meaningful Internet Systems: OTM 2016 Conferences. OTM 2016. Lecture Notes in Computer Science(), vol 10033. Springer, Cham. https://doi.org/10.1007/978-3-319-48472-3_15”. While it might not be fully relevant, it should be clearly stated why it was not considered.
• Your approach seems to be highly related to the field of similarity search. I would recommend looking at it. Here is a survey for graph-based similarity search, which might be relevant for you. “Shimomura, L. C., Oyamada, R. S., Vieira, M. R., & Kaster, D. S. (2021). A survey on graph-based methods for similarity searches in metric spaces. Information Systems, 95, 101507.”
• Additionally, the similarity search methods might interest you to optimize your approach.

Methods are replicable
• The submission of both code and data in multiple stages of processing is highly appreciated. And I would also encourage it in future submissions.
• I would recommend being more specific on the feature extraction.
• I am not sure about experiment 2, the methodology 2nd step. Does it mean that SAMOS only sees a subset of the models? It should be more clearly explained to avoid doubts about fairness.

Validity of the findings

First and foremost, I highly appreciate the release of data and code. However, there are a few weak points regarding the conclusions and discussion of the results. This is important for two reasons. First, to make a clear distinction regarding the novelty of your approach. And second, to provide deeper insight into the clone detection area. This should include explicit discussions of the impact of the decision taken in the design of your approach compared to state of the art.
• A further discussion regarding the choice of thresholds for Apromore (experiment 1) would be beneficial. The possible issue is the threat of disadvantaging the tool in comparison.
• I would suggest including the full model scope in experiment 3 for further validation.

Impact and novelty
• While largely present at the end of the paper, the novelty should be more explicitly stated. Namely, the principal differences between the baseline SAMOS framework and Appromore. In other words, what is and what is not considered by basic SAMOS/Appromore and the BPMN extension, supported by the experiments.

Data provided
• The data is provided, including the intermediate stages of the experiments.
• However, visualizations of the data set or experiments would be valuable. For example, visualization of the data clusters or statistical visualizations of the data set.

Conclusions well stated
• In experiment 1, you stated the challenge of choosing an appropriate similarity metric. However, insights into the behaviour of the metrics based on the experiment are not reflected in conclusion.
• The discussion on the improvements is rather vague and general. More concrete examples would be beneficial.
• The description of the differences and their impact should go beyond statements about higher precision and recall.
• In the discussion, lines 755-757, you claim that the problem with nested clones was addressed in the experimental evaluation. However, I did not find it there.

Additional comments

In summary, I think you present a good idea with well-thought experiments. However, the following points must be addressed:
1. Language
2. Structure of the text
3. Establishment of the goals and context
4. Further explanation of the reasoning behind decisions
5. Visualization of the data
6. Conclusion and discussion beyond precision and recall

---

## Round 0.2 · accepted · Accept

The paper can be accepted. Congratulations.

Reviewer 1 ·

Basic reporting

I thank the reviewers for answering the comments I had in the previous review. All the raised issued have been carefully addressed. Moreover, it is evident that the English in the paper is much better compared to the previous version. Overall, the paper is now substantially improved, and thus I happily recommend the acceptance.

Experimental design

The experimental design is sound.

Validity of the findings

Conclusions are supported with empirical evidence.

Additional comments

The paper has been substantially improved and it is publishable.